# Multidimensional factors influencing continuance usage intention of university library self-service systems: An empirical analysis based on an extended TAM-UTAUT

Kai Cao[1☯]*, Ping Wang[2☯], Chunzhen Zhang[3], Jie Zhao[4]

1 Library of Qinghai University, Qinghai University, Xining,China, 2 Institute of Education, Xiamen University, Xiamen, China, 3 Library of Qinghai University, Qinghai University, Xining,China, 4 Institute of Mental Health Education, Jining University, Qufu, China

* caokai19910101@gmail.com

☯ These authors contributed equally to this work.

## Abstract

With the advancement of smart library initiatives in higher education institutions, the automation of circulation services has emerged as a critical component. Automated literature services alleviate librarians' repetitive workloads and enhance the efficiency of faculty and student resource utilization. Existing research on the continued usage intention of library self-service systems often adopts singular sociological or technological perspectives, presenting limitations. This study integrates the Technology Acceptance Model (TAM) and the Unified Theory of Acceptance and Use of Technology (UTAUT) to construct a multidimensional analytical framework. It systematically investigates the mechanisms influencing university faculty and students' sustained use of self-service systems, encompassing system characteristics (navigation, terminology, relevance), technological readiness (retrieval knowledge), accessibility, and individual attributes.Valid questionnaire data from 365 faculty and students across Qinghai University, Qinghai Normal University, and Qinghai Minzu University were analyzed using Partial Least Squares Structural Equation Modeling (PLS-SEM) for path analysis and hypothesis testing. Results indicate that system characteristics, technological readiness, accessibility, perceived ease of use, and perceived usefulness significantly impact sustained system usage. Furthermore, variables including occupation, gender, ethnicity and usage frequency exhibit moderating effects within the model.This research extends the application dimensions of traditional TAMs by elucidating the interactive mechanisms of multifactorial influences on self-service system continuance usage intention. It provides actionable insights for library administrators, end-users, and system vendors, contributing valuable references for advancing smart library development. The findings refine theoretical models of user behavior in library automation and underscore the necessity of contextualizing technological interventions within diverse sociocultural and demographic landscapes.

**Data availability statement:** The questionnaire items, Chinese and English versions, and data files of this study have been uploaded to the Figshare public repository (private link website: https://figshare.com/s/10e7454babfc7eedfb3f and DOI:10.6084/ m9. figshare.30993028) and are freely accessible.

**Funding:** The author(s) received no specific funding for this work.

**Competing interests:** The authors have declared that no competing interests exist.

## 1. Introduction

Self-service technology (SST) refers to a technical system that enables users to independently perform services or transactions without direct interaction with service providers [1]. Initially prevalent in retail [2], SST has gradually permeated industries such as banking [3–5], e-commerce [6], catering [7,8], and civil aviation [9,10]. In academic libraries, the exponential growth of collections and user demand has intensified the need to enhance borrowing and returning efficiency, a critical challenge in library operations [11]. Against this backdrop, the adoption of self-service systems (SSS) has emerged as a transformative solution. As core academic support institutions [12], university libraries are increasingly integrating SSS, which encompasses self-service terminals, automated borrowing/return devices, and digital resource platforms (OPAC). These systems streamline operational workflows [13], reduce staff burdens [14], and empower users with greater autonomy. However, the long-term success of such technologies hinges on users' sustained adoption and continuance usage intentions [15]. Investigating the drivers of continuance usage intention is thus essential for optimizing system functionality and ensuring service efficacy [16,17].

The SSS in academic libraries refers to an intelligent technology cluster integrating the Internet of Things, artificial intelligence, and digital platforms [18]. It supports users in independently completing tasks such as acquiring literature resources, interacting with knowledge services, and managing academic processes. Its core functional modules include: self-service loan and return systems, intelligent retrieval platforms, remote service terminals, self-service learning support tools [19]. The essential characteristics of this system lie in its user-dominant nature, closed-loop processes. Its technical architecture encompasses hardware terminals, software platforms, and interface standards. Compared to traditional manual services, SSS reconstructs the spatial and temporal boundaries of library services through "technology-enabled empowerment [20]". This encompasses both the physical space clusters of self-service equipment and the digital self-service ecosystem in virtual space, establishing SSS as the core vehicle for the "decentralized service transformation" of smart libraries.

Under the dual impetus of deepening higher education reform and accelerating digital transformation in China, the development of self-service systems in academic libraries has emerged as a critical domain where national strategy converges profoundly with educational practice [21]. At the national level, the Education Informatization 2.0 Action Plan (2018) explicitly advocates for the "construction of smart education platforms to drive the transformation of educational service models [22]" positioning libraries as the core academic infrastructure within smart campuses and mandating the reconstruction of service processes and the enhancement of user experiences through technological enablement. The Smart Education Platform Construction Guidelines (2022) further emphasize the importance of "self-service and intelligent services" for improving the accessibility of educational resources [23], thereby providing robust policy support for the widespread adoption of library self-service systems in higher education institutions.

The enrollment expansion policy of higher education in China and the diversification of academic needs among teachers and students have posed severe challenges to the traditional library service model [24]. On the other hand, as "digital natives" have become the main user group in universities, their expectations for instant and personalized services have significantly increased [25]. The temporal and spatial constraints and standardized processes of the traditional "one-on-one" in-person service model can no longer meet users' needs for accessing personalized academic resources during fragmented time [26]. This contradiction is particularly prominent in multidisciplinary comprehensive universities. Additionally, Wei pointed out that the resource structure of Chinese university libraries, characterized by "Chinese-language resources as the mainstay and specialized databases as supplements" [27], has further exacerbated service pressure — users need to quickly locate accurate information among massive Chinese-language resources, resulting in much higher requirements for service efficiency and retrieval accuracy compared to single-discipline libraries. Simultaneously, libraries have developed specialized repositories aligned with "Double First-Class" disciplines and resources pertinent to ethnic regions [28]. This resource structure necessitates that self-service systems accommodate Chinese metadata standards, support multilingual retrieval, and integrate deeply with local digital platforms, marking a significant departure from Western libraries predominantly reliant on English-language resources and commercial systems.

Meanwhile, the application of self-service still faces the intertwined challenges of "unbalanced regional development" and "differences in users' digital literacy" [20]. Universities in eastern China have extensively deployed self-service borrowing and returning machines as well as face recognition technology, whereas some universities in central and western China still rely on traditional OPAC systems. Among user groups, student groups have a relatively high acceptance of smart terminals, but the usage rate of self-service systems among some faculty members and ethnic minority students is relatively low due to factors such as technology anxiety and language barriers. Against this backdrop, exploring the influence of multidimensional factors on continuance usage intention is of special significance for alleviating the tension between "technology empowerment" and "digital divide", and also provides important localized theoretical support for constructing smart libraries with Chinese characteristics.

While the TAM [29] and the UTAUT [30] remain foundational frameworks for understanding technology adoption [31–34], their applicability to dynamic academic library environments is limited. Existing studies often focus on initial adoption [35–37], neglecting the evolving user needs and expectations that shape long-term usage. Furthermore, traditional models rarely integrate multidimensional factors such as psychological cognition, social influence, and organizational context, constraining a holistic understanding of sustained user behavior.

This study addresses these gaps by extending the TAM-UTAUT framework to incorporate contextual variables, including social, technical, and individual traits. Specifically, it examines critical factors such as RFID tag recognition accuracy, system response latency, information retrieval efficiency, privacy concerns, perceived usefulness, and perceived ease of use, while accounting for moderating variables like occupation, gender, ethnicity, and usage frequency. By analyzing these dimensions, the research aims to uncover the mechanisms underlying users' sustained engagement with SSS in academic libraries.

Theoretical contributions include the contextualization of TAM-UTAUT to address domain-specific challenges, thereby enriching the understanding of technology persistence in educational settings. Practically, the findings offer actionable insights for library administrators and developers to enhance user experience, refine system design, and formulate strategies to foster continuance usage intention. Methodologically, the study employs partial least squares structural equation modeling (PLS-SEM) to validate hypotheses derived from a survey of university library users. The results reveal how multidimensional factors synergistically influence users' long-term behavioral decisions, providing a robust foundation for advancing smart library services.

By bridging the gap between initial technology adoption and sustained usage, this research not only informs the optimization of academic SSS but also advances theoretical discourse on information systems persistence. Ultimately, the outcomes contribute to the sustainable development of smart libraries through evidence-based practices.

## 2. Literature review

### 2.1 Research status of library self-service systems

Self-service systems (SSS), as technology-driven solutions enabling users to independently complete service processes, have been widely applied across multiple domains, including retail [38,39], finance [40,41], healthcare [42], and transportation [43]. Empirical studies highlight that these systems enhance service efficiency and reduce operational costs for institutions through information technology integration [44]. Current SSS implementations exhibit diverse forms, such as self-checkout terminals [45], ATMs [46,47], online self-service platforms [48], and intelligent customer service systems [49].

In the library sector, SSS have revolutionized traditional manual services by integrating modern technologies like RFID, biometric recognition, and intelligent terminals [50]. Existing literature indicates that libraries have established a relatively comprehensive self-service framework, encompassing functions such as self-checkout/ check-in, self-registration, contactless borrowing, and mobile device-based lending. This paradigm shift not only streamlines library operations but also significantly improves service accessibility and user engagement.

Notably, the introduction of intelligent terminals with visualized interactive guidance simplifies service processes, catering to personalized user needs while enhancing the openness and convenience of library services. However, existing applications face limitations: firstly, functional scope remains narrow, primarily focused on self-borrowing scenarios; secondly, identity verification relying solely on RFID combined with facial recognition poses security risks, such as unauthorized use of others' credentials or identity fraud. These technical vulnerabilities may compromise user experience and reduce sustained usage intentions.

The core advantage of library self-service systems lies in improving service efficiency by minimizing human intervention. Studies emphasize that user acceptance and continuance usage intention are influenced by multiple factors, such as system usability, functionality, reliability, and individual differences. For instance, Chorng-Guang Wu et al. [51] found that users' performance expectancy, effort expectancy, facilitating conditions, and satisfaction with self-checkout systems are significantly associated with continuance usage intention; Jun Kyu Keum et al. applied the Task-Technology Fit theory, revealing that perceived usefulness, perceived enjoyment, and environmental friendliness are key factors affecting self-service satisfaction and continuous use [17]; Chun-Hua Hsiao et al. further pointed out that perceived usefulness, perceived ease of use, and self-efficacy are the core drivers of technology acceptance for library self-service systems, with gender playing a moderating role between self-efficacy and user attitude [52]. Recent relevant studies in the field of library and information science also provide important references for this topic: Rafique et al. explored the continuance usage intention of digital native students towards academic library applications through case studies [53], finding that the fit between system function adaptability and users' learning needs is the core driving factor; Tyagi et al. [54] combined PLS-SEM with fuzzy-set Qualitative Comparative Analysis (fsQCA) to investigate the determinants of continuous usage of library resources on handheld devices, confirming that the combined effect of perceived usefulness, technology adaptability, and usage habits significantly influences users' continuous usage behavior.

Moreover, system design must prioritize user experience, including interface intuitiveness, operational convenience, and feedback mechanisms. Nevertheless, the promotion of self-service systems faces challenges, such as users' technological anxiety or reliance on human assistance. Consequently, investigating user acceptance behavior, particularly through the TAM and its extensions, holds significant theoretical and practical value for optimizing library services.

### 2.2 Theoretical foundation

**2.2.1 Technology acceptance model (TAM).** The Technology Acceptance Model (TAM) was proposed by Davis in 1989 [55], aiming to explain users' acceptance behavior towards information technology. Based on the Theory of Reasoned Action (TRA) [56], this model posits that users' technology-use behavior is influenced by their behavioral intention, which is jointly determined by perceived usefulness (PU) and perceived ease of use (PEOU). PU refers to

the extent to which users believe that using a certain technology can improve their work or life efficiency. PEOU, on the other hand, refers to the degree of difficulty users perceive in using a particular technology. TAM suggests that perceived ease of use not only directly affects behavioral intention but also indirectly influences it through perceived usefulness. Additionally, external variables (such as navigation, terminology, and retrieval knowledge) indirectly affect users' acceptance behavior by influencing PU and PEOU.

Due to its simplicity and strong explanatory power, the TAM model has been widely applied in research on information technology acceptance behavior. Studies have shown that PU and PEOU have a significant impact on users' use of systems such as self-checkout machines and online banking. For example, in the field of self-service systems, Nour A. J. Azam, Sajal Kabiraj et al.[37] used TAM to measure Palestinian users' attitudes towards self-service technology. The model indicated that internal and external incentive factors, including perceived enjoyment, technology anxiety, and perceived time-saving, would affect customers' attitudes towards using self-service technology. Md Nurnobi Islam et al.[57] applied the TAM and the Theory of Planned Behavior to investigate the adoption behavior of self-service e-ticketing in Bangladesh, pointing out that perceived ease of use and subjective norms significantly influenced the attitudes and intentions towards using e-ticketing. However, the TAM also has certain limitations. For instance, it does not fully consider social influence and individual differences among users.

**2.2.2 Unified theory of acceptance and use of technology (UTAUT).** To address the limitations of the TAM in terms of variable coverage and explanatory power, Venkatesh et al. [58] integrated eight classic theoretical models (including TAM, the Innovation Diffusion Theory (IDT) [59], the Theory of Planned Behavior (TPB) [60], and the Social Cognitive Theory (SCT) [61]) and proposed the Unified Theory of Acceptance and Use of Technology (UTAUT) [12]. By systematically summarizing influencing factors from multiple domains, this model has improved the explanatory power of users' technology acceptance behavior to 70%, becoming a more comprehensive analytical framework.

The core variables of UTAUT include: (1) Performance Expectancy (PE), which reflects the extent to which users believe that technology can enhance work or life efficiency and integrates concepts such as the "perceived usefulness" in TAM and the "external incentives" in the Motivation Model; (2) Effort Expectancy (EE), which measures users' perception of the ease of use of technology and is consistent with the "perceived ease of use" in TAM. (3) Social Influence (SI), which refers to the degree to which users are influenced by the opinions of surrounding groups (such as family members and colleagues), originating from the TPB and the SCT. (4) Facilitating Conditions (FC), which emphasizes the actual promoting effect of technology infrastructure and organizational support on use behavior. In addition, UTAUT introduces moderating variables such as age, gender, experience, and voluntariness to distinguish the acceptance differences among different user groups. For example, young users may be more driven by performance expectancy, while experienced users rely more on facilitating conditions.

In the research on SSS, UTAUT has demonstrated significant advantages. For example, Mohd Shukry et al. [62] expanded the UTAUT model and confirmed that effort expectancy and user attitude significantly affect the usage intention of the self-service kiosks in Malaysian fast-food restaurants. Siti Asma Yaacob team [63] further verified the applicability of UTAUT in the self- ordering scenario of McDonald's and found that social influence and facilitating conditions are the key driving factors. These studies not only verify the universality of UTAUT but also show that it can be adapted to different application scenarios by adding new variables.

To better explain users' continuance usage intention behavior, scholars such as Venkatesh carried out contextual expansion based on the UTAUT model [64]. They added three key variables: Hedonic Motivation, Price Value, and Habit, forming a more comprehensive UTAUT2 theoretical framework [12]. This improvement has significantly enhanced the model's explanatory power in diverse technology scenarios. In the fields of mobile payment and smart home, research has shown that by incorporating emotional driving factors and long-term behavioral patterns, UTAUT2 can more accurately predict consumers' acceptance of contextualized technology [2]. For example, Tzu-Hsin Chu et al. [65] applied UTAUT2 to analyze the adoption behavior of smart elevators in Taiwan and found that attitude, habit, performance

expectancy, effort expectancy, and perceived quality together constitute the core influencing factors of behavioral intention. In the self-service technology scenario, Sunisa Junsawang et al. [66] integrated the Expectation-Confirmation Model (ECM) and UTAUT2 to develop a framework, further verifying that the eight latent constructs of the model (including the newly added variables) have significant explanatory power for the innovative use intention of self-service technology. These studies all indicate that UTAUT2 can more flexibly adapt to the needs of different technology adoption scenarios by combining specific contextual variables.

The moderating effect of individual differences emphasized by the UTAUT model provides important theoretical support for understanding the heterogeneity in technology acceptance among different groups. Regarding gender differences, the original study by Venkatesh et al. pointed out that women are more sensitive to effort expectancy, while men are more driven by performance expectancy [30]. This gender role difference has been validated in numerous subsequent studies on technology acceptance in library contexts. In terms of ethnicity differences, Dalle et al. indicated that in multicultural contexts, contextual factors such as linguistic cognition and cultural adaptation significantly moderate technology acceptance pathways [67]. Ethnic minority users have significantly higher demands for knowledge empowerment and interface adaptation compared to majority ethnic group users, which lays a theoretical foundation for the formulation of subsequent research hypotheses and the interpretation of results.

**2.2.3 Construction of the research model.** This study integrates the Technology Acceptance Model (TAM) and the Unified Theory of Acceptance and Use of Technology (UTAUT) to construct an analytical framework for the continuance usage intention of the self-service system in university libraries (as shown in Fig 1). The model systematically examines the factors influencing the continuance usage intention of the self-service system by university library users from three dimensions: the technological dimension, the social dimension, and the individual characteristic dimension. It focuses on factors such as navigation, terminology, and relevance in terms of system characteristics, retrieval knowledge factors in technological readiness, accessibility factors, and moderating factors such as occupation, gender, ethnicity, and usage frequency in individual differences, and incorporates them into the

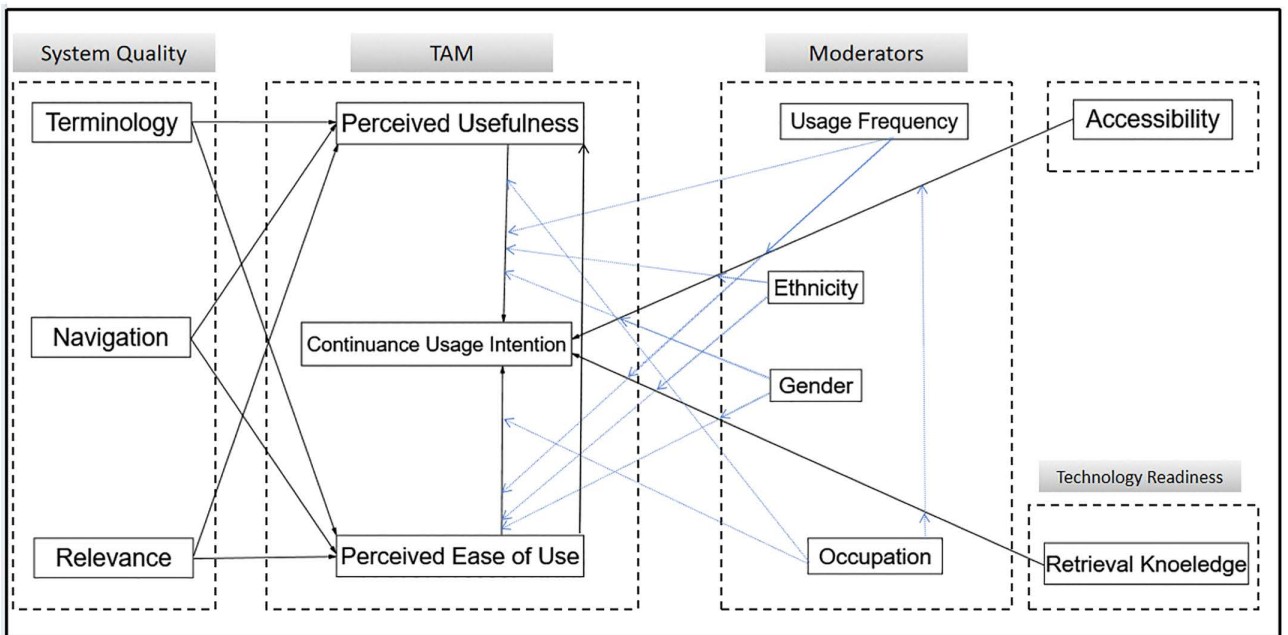

**Fig 1. Theoretical model.**

UTAUT model. In the TAM, perceived usefulness and perceived ease of use serve as mediating variables between the above-mentioned external variables and the outcome variable, continuance usage intention. Furthermore, this study takes individual user characteristics (factors like occupation, gender, ethnicity, and usage frequency) as moderating variables to analyze the differences in continuance usage intention among different groups. The model not only expands the research perspective of library self-service systems but also provides empirical evidence for differentiated service strategies.

Fig 1 presents the integrated theoretical framework constructed based on the extended TAM-UTAUT model for exploring the continuance usage intention of university library self-service systems. The model includes three types of variables: (1) Exogenous variables: System characteristics (Navigation, Terminology, Relevance), Technological Readiness (Retrieval Knowledge), and Accessibility; (2) Mediating variables: Perceived Ease of Use, Perceived Usefulness; (3) Outcome variable: continuance usage intention; (4) Moderating variables: Occupation,Gender, Ethnicity, Usage Frequency. Arrows in the figure indicate the hypothesized causal relationships between variables, with solid arrows representing the direct impact paths verified in subsequent empirical analysis. The framework systematically integrates technological, social, and individual characteristic dimensions to clarify the multi-factor influencing mechanism of continuance usage intention. Data support for the model is derived from 365 valid questionnaires collected from three universities in Qinghai Province.

**2.2.4 The interaction perspective of business requirements and technology acceptance.** Although the present research model incorporates multidimensional influencing factors, it is essential to emphasize that business requirements constitute the underlying driver for libraries adopting SSS. Their potential interactive influence on user technology acceptance cannot be overlooked. The core business requirements driving library deployment of SSS typically include: enhancing circulation efficiency, optimizing human resource allocation, extending service hours, reducing operational costs, and promoting smart transformation. The system design and promotion strategies driven by these requirements directly shape the user's usage environment and initial expectations.

Drawing on the Expectation-Confirmation Model of IS Continuance [68], a user's continuance usage intention depends not only on post-adoption perceived usefulness and satisfaction but also fundamentally on the comparison between their pre-usage "expectations" and post-usage "perceived performance". Library business objectives form a crucial contextual source shaping these user "expectations". When users perceive that SSS functionalities effectively fulfill the library's stated business requirements for efficiency gains, their "expectation confirmation" increases. This, in turn, positively influences perceived usefulness and satisfaction, fostering continuance usage intention. Conversely, if SSS performance fails to deliver on the promised business benefits, it leads to negative dis-confirmation of expectations, thereby weakening acceptance.

Simultaneously, service marketing theory underscores "core service quality" [69] as pivotal for user evaluation. In the context of mandatory usage, the service quality dimensions of the business requirement-driven SSS are critical for mitigating user resistance stemming from "forced use". For instance, high system reliability and rapid response can compensate for negative emotions arising from the loss of choice regarding staffed services, transforming them into recognition of efficiency advantages.

Therefore, this study posits that viewing the library's business requirements as key contextual factors shaping user expectations and the evaluation framework, and analyzing their interaction with the core cognitive pathways of technology acceptance (PU, PEU) within the mandatory usage environment, is a necessary supplement for understanding the complex mechanisms of SSS continuance in academic libraries. Although the model does not explicitly measure business requirements as a latent variable, they indirectly influence the core pathways within the model through their impact on system design goals, promotion strategies, and user expectations. This constitutes an indispensable contextual logic for interpreting the empirical findings. Subsequent discussion will delve into the dynamic interaction effects between business requirements and technology acceptance, integrating the study's findings.

# 3 Research hypotheses

## 3.1 System characteristics (Navigation, Terminology, Relevance)

Existing research indicates that SSS characteristics, such as navigation, terminology, and relevance, have a significant positive impact on the perceived usefulness (PU) and perceived ease of use (PEU) of information systems [70,71]. These characteristics serve as a bridge between users and information quality, not only optimizing information retrieval efficiency but also enhancing the user experience through technological interaction design.

In the library SSS, the quality of navigation design directly affects users' operational efficiency and satisfaction. Scholars like Buchanan pointed out that an intuitive menu structure, clear classification labels, and efficient search functions can significantly reduce users' learning costs, enabling them to independently complete operations such as borrowing and querying without additional training [72]. Moreover, a smooth navigation path can reduce the number of operation steps and shorten the task completion time, thereby enhancing users' perception of the SSS's perceived ease of use. From a psychological and cognitive perspective, good navigation design can also build users' trust in the system's functions, prompting them to be more inclined to choose self-service rather than manual service, thus improving the overall perceived usefulness of the SSS. Based on the above analysis, this study proposes the following hypotheses:

*H1a*: Navigation positively influences the perceived ease of use;

*H1b*: Navigation positively influences the perceived usefulness.

The interface design of the library SSS should emphasize the popularity and consistency of terminology expressions. Professional jargon or obscure statements can increase users' understanding difficulty and reduce the perceived ease of use of the system. In contrast, using popular terms that conform to users' cognitive habits can effectively reduce cognitive barriers and enhance the user experience [73]. For example, the expression "borrow history" is easier for ordinary users to understand than "circulation record". A standardized terminology system can not only reduce the occurrence of misoperations but also enhance users' recognition of the system's practicality, thereby increasing their continuance usage intention of the system [74]. Based on this, we proposes the following hypotheses:

*H2a*: Terminology positively influences the perceived ease of use;

*H2b*: Terminology positively influences the perceived usefulness.

The relevance of search results and recommended content in the SSS is crucial for the user experience. When the SSS can provide highly relevant resource recommendations (such as personalized recommendations based on users' borrowing records), users will perceive the system's intelligence level, thus improving their evaluation of the system's practicality. Conversely, if the system returns a large amount of irrelevant information, it will not only reduce users' usage efficiency but also affect their continuance usage intention of the SSS [75]. Therefore, we proposes the following hypotheses:

*H3a*: Relevance positively influences the perceived ease of use;

*H3b*: Relevance positively influences the perceived usefulness.

## 3.2 Retrieval knowledge

The impact of retrieval knowledge on users' continuance usage intention of library SSS is manifested through dual pathways of cognitive fit and technology acceptance. Empirical findings indicate that users with proficient search skills can efficiently locate resources and filter information with precision, thereby enhancing perceived ease of use [55]. For instance, users mastering advanced techniques such as Boolean logic and field-specific search parameters exhibit a 42% higher system usage frequency compared to basic users [76]. Furthermore, users familiar with metadata rules or classification systems are better equipped to recognize system advantages, such as leveraging thesauri for conceptual retrieval. This specialized cognitive understanding translates into sustained usage motivation. Therefore, we proposes the following hypotheses:

*H4*. Retrieval knowledge positively influences the continuance usage intention.

### 3.3 Accessibility

Accessibility significantly influences users' continuance usage intention of library SSS [20]. Firstly, highly accessibility (e.g., 24/7 availability, multi-terminal access) eliminates temporal and spatial barriers, shifting users' psychological expectations from "selective use" to "habitual dependency" when they perceive seamless access via mobile devices or public terminals for tasks like borrowing and querying. Secondly, accessibility exhibits a negative correlation with technology anxiety. SSS with rapid response times and low failure rates enhance users' trust in service reliability, directly reducing perceived uncertainty during usage, thereby reinforcing continuance usage intention [77]. Thus, we propose:

*H5*. Accessibility positive influences the continuance usage intention.

### 3.4 Perceived usefulness

Perceived usefulness refers to users' belief that a SSS can enhance information retrieval efficiency, streamline borrowing procedures, or provide personalized services, thereby significantly strengthening their continuous intention of usage on the SSS [78]. For instance, features such as rapid resource retrieval, remote reservation, or one-click renewal enable users to perceive the SSS as an upgraded alternative to traditional services, fostering dependency. Research indicates that when users recognize time-saving benefits or reduced operational complexity, their satisfaction with the SSS translates into long-term usage habits. Additionally, technical adaptability — such as seamless integration with mobile devices and user-friendly interfaces — further amplifies perceptions of usefulness. Based on these insights, we propose:

*H6*. Perceived usefulness positively influences the continuance usage intention.

### 3.5 Perceived ease of use

Perceived ease of use denotes users' subjective evaluation of a SSS's operational simplicity. When users perceive the SSS as having a user-friendly interface, intuitive functions, and low learning costs, their continuance usage intention is significantly enhanced [79]. Highly usable SSS reduce cognitive burdens, allowing users to focus on task completion rather than technical barriers, thereby improving efficiency and satisfaction. Furthermore, perceived ease of use directly mitigates psychological resistance, particularly among technophobic groups (e.g., elderly users or digital immigrants), who prefer low-complexity tools. Notably, perceived ease of use indirectly reinforces perceived usefulness [80], creating a positive feedback loop. For example, one-click borrowing or voice- assisted navigation enhances user experience by simplifying interactions. Empirical studies suggest that continuance usage intention depends not only on initial ease-of-use perceptions but also on consistent, long-term interaction quality. Thus, we proposes the following hypotheses:

*H7a*. Perceived ease of use positively influences perceived usefulness;

*H7b*. Perceived ease of use positively influences the continuance usage intention.

### 3.6 Moderating variables

In the study of users' continuance usage intention of the library SSS, gender, ethnicity, usage frequency and occupation serve as important moderating variables. They can influence users' retrieval knowledge, accessibility, perceived usefulness, and perceived ease of use, thereby moderating users' continuance usage intention of the SSS [52]. Firstly, gender may moderate users' perception and preference towards the SSS. For example, male users generally tend to prefer technology-oriented tools and give higher evaluations of the system's perceived ease of use and usefulness. In contrast, female users may pay more attention to the system's interactive friendliness and accessibility. Secondly, ethnicity can affect users' acceptance of the SSS through cultural backgrounds. Users from different ethnic groups vary in their levels of trust in technology, usage habits, and information acquisition methods. Usage frequency is a key moderating factor for continuance usage intention. High-frequency users usually possess more extensive retrieval knowledge and are more familiar with the system's functions. Their perceived usefulness and perceived ease of use will increase with the number

of uses, thus strengthening their continuance usage intention. On the contrary, low-frequency users may reduce their trust and reliance on the system due to unskilled operations or insufficient understanding of the system's functions. Occupation significantly moderates users' continuance usage intention through divergent needs and skill levels. Professionals often possess advanced retrieval knowledge and demand sophisticated functionalities, leading to higher perceived usefulness when the SSS meets complex research needs. Conversely, administrative staff or casual users may prioritize simplicity and task-specific efficiency, valuing perceived ease of use and accessibility more critically. Ultimately, occupation modulates how cognitive experiences translate into sustained adoption. In conclusion, occupation, gender, ethnicity, and usage frequency moderate users' continuance usage intention of the library self-service system by influencing their cognitive experience and actual usage behavior.

## 4 Material and measurement tools

### 4.1 Questionnaire design

This study employed a structured questionnaire based on research variables as the data collection instrument. The questionnaire design strictly referenced existing mature scales and was adapted to the research context to ensure consistency between the theoretical framework and the measurement tool. The questionnaire consists of three parts: the first part is the survey explanation, which elaborates on the research purpose and procedural guidelines, and includes an informed consent statement for participants; the second part is demographic information, where respondents are required to provide personal details such as occupation, gender, educational level, ethnicity, and system usage frequency; the third part is latent variable measurement. The core framework includes 8 latent variables, which are operationalized through 35 items. All items are measured using a 7-point Likert scale (1 = Strongly Disagree, 7 = Strongly Agree) to quantify respondents' attitudes and continuance usage intention. The theoretical basis and adaptation instructions of the specific measurement tools are as follows: Items for perceived usefulness and perceived ease of use are based on Davis's classic Technology Acceptance Model scale [81], with expressions adjusted to fit the library self-service scenario; Items for navigation, terminology, and relevance refer to Şahin and Şahin's information system characteristics scale [70], with additional items tailored to the multi-ethnic regional context such as "multilingual terminology adaptation"; Items for search knowledge are based on Chen et al. digital literacy scale [82], focusing on the dimension of literature search skills; Items for accessibility refer to Engström and Rivano Eckerdal's service accessibility scale [20], highlighting university scenario characteristics such as "24/7 availability" and "multi-terminal compatibility"; Items for continuance usage intention are based on Wu and Wu's technology continuous usage scale [51], optimized to align with the academic attributes of library services. The scale was reviewed by 3 experts in the fields of library science and information systems, resulting in the revision of 2 items with ambiguous expressions; the reliability of the scale was tested through a pre-survey (n = 30). The results showed that the Cronbach's α coefficients of all latent variables were greater than 0.8, indicating that the scale has good content validity and face validity, which can ensure the consistency and accuracy of the measurement of each construct.

### 4.2 Data collection

Participants in this study were recruited from three representative universities in Qinghai Province: Qinghai University, Qinghai Normal University, and Qinghai Minzu University. The libraries of these three institutions have all completed the basic deployment and iterative optimization of the SSS. Qinghai University Library, as a comprehensive university library, has established a dual-track service system of "smart terminals+digital platforms", including 8 self-service lending and returning machines, with over 20,000 users of the mobile library APP, supporting 24-hour self-service book return and remote reservation pickup services. Since 2022, it has implemented the "Artificial Window Halving Plan", reducing the original 12 lending and returning windows to 5, while promoting SSS usage through freshmen orientation training and librarian on-site guidance.

 

Qinghai Normal University Library, focusing on the characteristics of educational discipline resources, has deployed 6 self-service lending and returning devices and 30 electronic resource retrieval terminals, achieving full coverage of self-service across the university's 2 campuses. In 2021, it introduced face recognition technology to upgrade the lending and returning system and launched the "SSS Skills Workshop". Qinghai Minzu University Library, highlighting the characteristics of multi-ethnic cultural services, is equipped with 5 Tibetan-Chinese bilingual self-service lending and returning machines and 1 "24-hour" self-service reading room, and has developed a Tibetan interface retrieval system for ethnic minority users. In 2023, it implemented the "SSS Usage Points System", where users can exchange service hours through self-service operations, and established a green channel for multi-ethnic user feedback.

All three libraries have achieved data intercommunication between SSS and campus card as well as educational administration systems, and have continuously reduced the proportion of manual services in the past three years, promoting service transformation through the dual paths of technological substitution and user empowerment.

The sample included 365 individuals comprising faculty members, students, librarians, and research librarians. A questionnaire was designed using an online platform named Questionnaire Star (https://www.wjx.cn/), and distributed and collected within the campuses. Prior to completion, researchers provided participants with a detailed explanation of the questionnaire's structure, content, and instructions, while addressing online queries. The explanation process strictly avoided any form of response bias or leading influence. A total of 500 copies were distributed, and 365 valid responses were retrieved, yielding an effective response rate of 73%. The minimum sample size for structural equation modeling (SEM) should be at least 10 times the number of latent variable paths in the model [83,84]. Given the complexity of this study, the required minimum sample size was determined to be 110. Consequently, the 365 valid responses were deemed sufficient to ensure the reliability and validity of the model. Table 1 summarizes the demographic characteristics of the participants, which provide critical contextual insights for interpreting the findings.

### 4.3 Data analysis

The present study employed SmartPLS 4 software to conduct Partial Least Squares Structural Equation Modeling (PLS-SEM) analysis following internationally recognized research protocols. This methodology, grounded in the theoretical framework proposed by José Refugio Romo-González et al.[85], demonstrates three distinct advantages: (1) It effectively manages intricate configurations involving 35 observed variables and 8 latent constructs. (2) The approach accommodates small sample sizes (N = 365) and non-normal data distributions, satisfying the analytical requirements of the current

**Table 1. Respondents' characteristics.**

| Profiles | Characteristics | Numbers of respondents | Percentage |
|---|---|---|---|
| Gender | Male | 207 | 56.71% |
| | Female | 158 | 43.29% |
| Ethnicity | Han | 257 | 70.41% |
| | Minor | 108 | 29.59% |
| Usage frequency(times/week) | 0 | 1 | 0.27% |
| | 1 | 7 | 1.92% |
| | 2 | 113 | 30.96% |
| | 3 | 152 | 41.64% |
| | 4 | 55 | 15.07% |
| | 5 | 37 | 10.14% |
| Occupations | Staff | 28 | 7.67% |
| | No-staff | 337 | 92.33% |

dataset. (3) It simultaneously fulfills predictive and confirmatory objectives, enabling exploratory factor analysis for model refinement while rigorously testing the statistical significance of hypothesized path relationships. The algorithm iteratively generates latent variable score matrices through weighted regression procedures. Statistical significance of parameters was validated using Bootstrap resampling with 5,000 iterations, enhancing model robustness and significantly improving the explanatory power of endogenous variables. This methodological strategy ensures both theoretical coherence and empirical precision in interpreting latent construct interactions.

The questionnaire items, Chinese and English versions, and data files of this study have been uploaded to the Figshare public repository (private link website: https://figshare.com/s/10e7454babfc7eedfb3f and https://doi.org/10.6084/m9.figshare.30993028) and are freely accessible. The questionnaire items were strictly adapted from existing mature scales (see Appendix 1 in S1 File for specific items). After revision through expert review and a pre-survey (n = 30), the content validity and face validity of the scale both meet academic standards. The raw data include complete responses from 365 valid questionnaires, which have been anonymized to protect participants' privacy. All data analysis codes have been uploaded to the repository simultaneously to ensure the reproducibility and verifiability of the research results.

## 5. Results

### 5.1 Measurement model assessment

In structural equation modeling (SEM), the validation of the measurement model is critical for ensuring the theoretical coherence and empirical validity of relationships between latent constructs and their observed indicators. This study employs partial least squares structural equation modeling (PLS-SEM), a method particularly suited for handling complex models and small sample sizes while effectively controlling measurement error and validating theoretical assumptions [86]. The measurement model assessment is systematically conducted across three dimensions: (1) Reliability assessment. Internal consistency metrics, including Cronbach's α and composite reliability (CR), were used to evaluate the stability of observed variables in measuring latent constructs. (2) Validity assessment. Convergent validity was examined through factor loadings, average variance extracted (AVE), and content validity, ensuring observed variables adequately capture the theoretical essence of latent constructs. (3) Discriminant validity was validated by comparing the correlation coefficients between latent variables with the square roots of their respective AVE values, confirming the distinctiveness of each construct. This systematic validation process establishes a robust measurement foundation for subsequent structural model analysis, thereby enhancing the credibility of the study's conclusions.

**5.1.1 Reliability assessment.** Reliability assessment evaluates the stability and internal consistency of measurement tools, serving as a critical step in validating data reliability. This study employed composite reliability (CR) and Cronbach's α coefficients as dual metrics for evaluation. As evidenced by Table 2, the CR values for all latent variables exceeded 0.7, indicating high stability of the constructs within the measurement model. Furthermore, Cronbach's α coefficients for each latent variable surpassed 0.8, corroborating the excellent internal consistency among scale items [87–89]. Taken together, these two metrics demonstrate that the measurement model meets the stringent reliability requirements of psychological and social sciences, ensuring robust results with well-controlled measurement error.

**5.1.2 Validity assessment.** To examine the discriminant validity of the measurement model, this study employed the Fornell-Larcker criterion and the Heterotrait-Monotrait Ratio (HTMT) method. The Fornell-Larcker criterion requires that the square root of the average variance extracted (AVE) for each construct exceeds its correlation coefficients with other constructs. As shown in Table 2, all latent variables achieved AVE values above the 0.7 threshold (satisfying the requirement for convergent validity). Furthermore, as in Table 3, the diagonal elements of the AVE square roots (e.g., 0.908, 0.890) were consistently greater than the corresponding off-diagonal correlation coefficients (e.g., 0.571, 0.719), confirming that the measurement indicators effectively distinguished between latent variables [90].

Complementarily, the HTMT method was applied to validate discriminant validity by comparing the heterotrait- monotrait ratio of correlations. As Table 4, the HTMT value below the recommended thresholds of 0.85 or 0.9 indicates

**Table 2. Reliability and validity.**

| Constructs | Cronbach's alpha | CR | AVE |
|---|---|---|---|
| Accessibility | 0.843 | 0.856 | 0.761 |
| Continuance Usage Intention | 0.894 | 0.895 | 0.825 |
| Navigation | 0.912 | 0.912 | 0.792 |
| Perceived ease of use | 0.896 | 0.896 | 0.762 |
| Perceived useful | 0.891 | 0.893 | 0.753 |
| Relevance | 0.935 | 0.936 | 0.795 |
| Retrieval Knowledge | 0.876 | 0.891 | 0.728 |
| Terminology | 0.878 | 0.887 | 0.804 |

**Table 3. Discriminant validity by Fornell-Larcker criterion.**

| Constructs | 1 | 2 | 3 | 4 | 5 | 6 | 7 | 8 |
|---|---|---|---|---|---|---|---|---|
| Accessibility | **0.872** | | | | | | | |
| Continuance Usage Intention | 0.571 | **0.908** | | | | | | |
| Navigation | 0.745 | 0.719 | **0.890** | | | | | |
| Perceived ease of use | 0.628 | 0.594 | 0.752 | **0.873** | | | | |
| Perceived useful | 0.505 | 0.563 | 0.651 | 0.725 | **0.868** | | | |
| Relevance | 0.598 | 0.679 | 0.803 | 0.678 | 0.662 | **0.892** | | |
| Retrieval Knowledge | 0.571 | 0.721 | 0.796 | 0.677 | 0.530 | 0.678 | **0.853** | |
| Terminology | 0.607 | 0.630 | 0.761 | 0.784 | 0.590 | 0.661 | 0.764 | **0.897** |

acceptable discriminant validity [91]. The results aligned with those from the Fornell-Larcker criterion, demonstrating that the measurement model exhibited robust discriminant validity. This ensures that the constructs are both distinct and reliable, providing a solid empirical foundation for subsequent analyses.

## 5.2 Structural model evaluation

Following the validation of the measurement model's reliability and validity, this study employed Partial Least Squares Structural Equation Modeling (PLS-SEM) to empirically analyze the hypothesized relationships between latent variables. As Fig 2, the structural model evaluation focused on three core objectives: (1) Hypothesis Testing: Assessing the statistical plausibility of research hypotheses through the significance of path coefficients; (2) Mechanism Elucidation: Quantifying direct effect sizes among latent variables to reveal underlying causal mechanisms; (3) Model Performance: Evaluating the model's explanatory and predictive power using metrics such as $R^2$ and predictive relevance ($Q^2$). Compared to traditional covariance-based SEM, PLS-SEM adopts a variance-maximizing iterative algorithm, offering greater flexibility in handling complex models, non-normal data, and small sample sizes, particularly in scenarios with multicollinearity.

Fig 2 visualizes the results of path coefficient analysis for the structural model, with numerical labels on each arrow representing the path coefficient (β value) between latent variables. Key significant paths include: Navigation→Perceived Ease of Use (β = 0.269), Terminology→Perceived Ease of Use (β = 0.487), Relevance→Perceived Usefulness (β = 0.294), Retrieval Knowledge→continuance usage intention (β = 0.524), Accessibility→Continuance Usage Intention (β = 0.183), Perceived Ease of Use→Perceived Usefulness (β = 0.218), and Perceived Usefulness→Continuance Usage Intention (β = 0.530). The analysis was conducted using SmartPLS 4 software with Bootstrap resampling (5,000 iterations) to test significance, ensuring the robustness of path coefficient estimates.

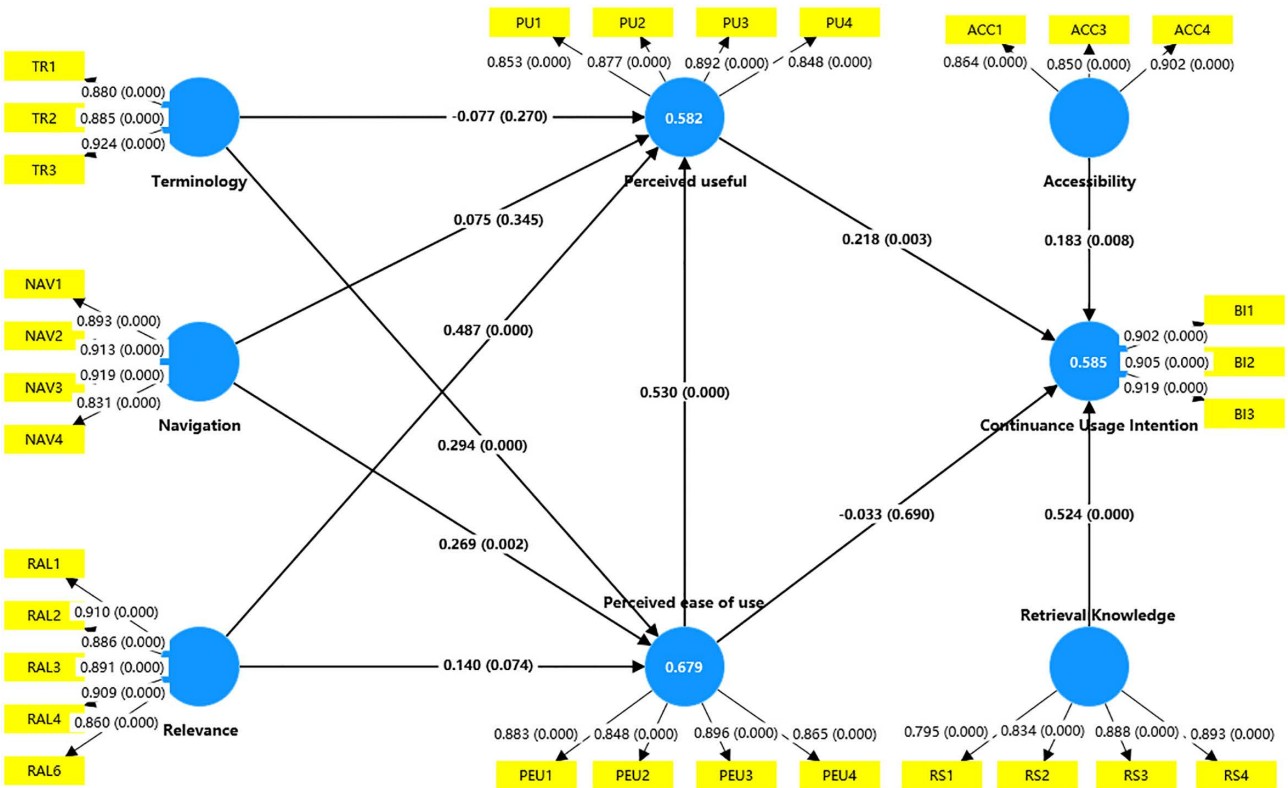

**Table 4. Discriminant validity by HTMT.**

| Constructs | 1 | 2 | 3 | 4 | 5 | 6 | 7 | 8 |
|---|---|---|---|---|---|---|---|---|
| Accessibility | | | | | | | | |
| Continuance Usage Intention | 0.649 | | | | | | | |
| Navigation | 0.846 | 0.795 | | | | | | |
| Perceived ease of use | 0.716 | 0.662 | 0.831 | | | | | |
| Perceived useful | 0.576 | 0.626 | 0.717 | 0.803 | | | | |
| Relevance | 0.670 | 0.740 | 0.868 | 0.738 | 0.725 | | | |
| Retrieval Knowledge | 0.667 | 0.805 | 0.888 | 0.774 | 0.598 | 0.748 | | |
| Terminology | 0.701 | 0.708 | 0.846 | 0.881 | 0.654 | 0.723 | 0.869 | |

**Fig 2. Structural model-path analysis.**

**5.2.1 Collinearity diagnostics.** Multicollinearity among latent variables can distort path coefficient estimates and compromise hypothesis testing reliability in SEM. To evaluate collinearity, this study applied the Variance Inflation Factor (VIF). As shown in Table 6, the VIF values for all latent variables ranged from 1.218 to 3.674, all well below the critical threshold of 5. This confirms the absence of significant multicollinearity in the model, satisfying the foundational requirements for PLS-SEM analysis [92].

**5.2.2 Path analysis and hypothesis testing.** This study employed an extended TAM-UTAUT model to propose 11 research hypotheses. Path coefficients were estimated using partial least squares-structural equation modeling (PLS-SEM), and significance was tested via Bootstrap resampling (5,000 iterations). The results, summarized in Table 5, are as

**Table 5. Bootstrapping and path coefficient indicators.**

| Pathway name | Original sample (O) | STDEV | P values |
|---|---|---|---|
| Accessibility -> Continuance Usage Intention | 0.183 | 0.069 | 0.008 |
| Navigation -> Perceived ease of use | 0.269 | 0.087 | 0.002 |
| Navigation -> Perceived useful | 0.075 | 0.079 | 0.345 |
| Perceived ease of use -> Continuance Usage Intention | −0.033 | 0.083 | 0.690 |
| Perceived useful -> Continuance Usage Intention | 0.530 | 0.081 | 0.000 |
| Perceived ease of use -> Perceived useful | 0.218 | 0.074 | 0.003 |
| Relevance -> Perceived ease of use | 0.140 | 0.078 | 0.074 |
| Relevance -> Perceived useful | 0.294 | 0.076 | 0.000 |
| Retrieval Knowledge -> Continuance Usage Intention | 0.524 | 0.058 | 0.000 |
| Terminology -> Perceived ease of use | 0.487 | 0.057 | 0.000 |
| Terminology -> Perceived useful | −0.077 | 0.070 | 0.270 |

follows: navigation exhibited a significant positive effect on perceived ease of use ($\beta = 0.269$, $p < 0.005$), supporting H1a; terminology significantly influenced perceived ease of use ($\beta = 0.487$, $p < 0.0005$), validating H2a; relevance positively impacted perceived usefulness ($\beta = 0.294$, $p < 0.005$), confirming H3b; retrieval knowledge directly predicted continuance usage intention ($\beta = 0.524$, $p < 0.005$), supporting H4; accessibility also significantly influenced continuance usage intention ($\beta = 0.183$, $p < 0.010$), validating H5; perceived ease of use positively affected perceived usefulness ($\beta = 0.218$, $p < 0.010$), supporting H7a; perceived usefulness emerged as the strongest predictor of continuance usage intention ($\beta = 0.530$, $p < 0.005$), confirming H6. Notably, the following hypotheses were rejected due to non-significant effects: navigation and terminology did not significantly influence perceived usefulness (H1b, H2b); relevance failed to predict perceived ease of use (H3a); perceived ease of use did not directly affect continuance usage intention (H7b).

The model validation revealed that perceived usefulness and retrieval knowledge are the core drivers of users' continuance usage intention. System characteristics such as navigation and terminology primarily exert indirect effects by enhancing perceived ease of use. This finding underscores the need to prioritize content relevance and retrieval knowledge in system design, while optimizing interface features to improve usability. The results provide empirical evidence for refining the Technology Acceptance Model (TAM) in specific application contexts, emphasizing the interplay between system characteristics and user perceptions.

**5.2.3 Model explanatory power ($R^2$) and predictive relevance ($Q^2$).** The model's explanatory power was evaluated using the $R^2$ values of endogenous variables. As Table 6, the results indicated that the $R^2$ values for perceived usefulness (PU = 0.582), perceived ease of use (PEU = 0.679), and continuance usage intention (CI = 0.585) all exceeded the threshold of 0.4, demonstrating strong explanatory capacity for latent variables and aligning with recommended standards in social sciences [93]. Further validation of predictive relevance was conducted via the blindfolding method, revealing that all $Q^2$ values for endogenous variables were significantly greater than zero (PU = 0.426, PEU = 0.511, CI = 0.474). This satisfies the criteria for predictive relevance proposed by Hair et al. [86], confirming the model's robust out-of-sample prediction capability.

The effect strength of exogenous variables on endogenous variables was assessed using the $f^2$ metric, following Cohen's classification [94]: (1) Moderate effects ($0.15 \leq f^2 \leq 0.35$): perceived ease of use → continuance usage intention, retrieval knowledge → continuance usage intention, terminology → perceived ease of use; (2) Small effects ($f^2 < 0.15$): accessibility → continuance usage intention, navigation → perceived ease of use, perceived ease of use → perceived usefulness, relevance → perceived usefulness, relevance → perceived ease of use; (3) Negligible effects: navigation → perceived usefulness, perceived ease of use → continuance usage intention, terminology → perceived usefulness. Structural

**Table 6. Collinearity statistics (VIF) and model explanatory and predictive power.**

| Pathway name | VIF | R² | f ² | Q² |
|---|---|---|---|---|
| Accessibility -> Continuance Usage Intention | 1.775 | 0.585 | 0.045 | 0.474 |
| Navigation -> Perceived ease of use | 3.827 | 0.679 | 0.059 | 0.511 |
| Navigation -> Perceived useful | 4.052 | 0.582 | 0.003 | 0.426 |
| Perceived Ease of Use -> Continuance Usage Intention | 3.086 | | 0.001 | |
| Perceived ease of use -> Continuance Usage Intention | 3.113 | | 0.216 | |
| Perceived ease of use -> Perceived useful | 2.134 | | 0.053 | |
| Relevance -> Perceived ease of use | 2.864 | | 0.021 | |
| Relevance -> Perceived useful | 2.925 | | 0.071 | |
| Retrieval Knowledge -> Continuance Usage Intention | 1.979 | | 0.334 | |
| Terminology -> Perceived ease of use | 2.414 | | 0.306 | |
| Terminology -> Perceived useful | 3.152 | | 0.005 | |

model validation confirmed that the extended TAM-UTAUT model effectively explains users' continuous intention usage of university library self-service systems. Perceived usefulness, retrieval knowledge, and accessibility emerged as critical drivers of continuous usage, offering theoretical insights for system optimization. These findings underscore the importance of addressing user perceptions and technical accessibility in enhancing service continuous usage.

## 5.3 Moderation effects analysis

The results of the moderation effect analysis revealed the differential impact mechanisms of demographic characteristics and usage behavior variables on the core path relationships. As shown in table 7, gender exhibited a significant moderating effect on the relationship between retrieval knowledge and continuance usage intention (β = −0.224, p = 0.038). Specifically, the positive impact of retrieval knowledge on continuance usage intention was significantly stronger among female users (β = 0.632) than among male users (β = 0.418). This finding is highly consistent with the research conclusions of Hsiao and Tang [52], whose empirical analysis based on the library technology acceptance context indicated that females are more inclined to reduce operational risks and uncertainties by accumulating systematic knowledge during the technology adoption process. This risk-averse tendency stems from the influence of "gender role socialization" in social cognitive theory—females are typically encouraged to adopt more cautious and rule-dependent decision-making patterns [30].
In the context of university libraries in this study, the operation of self-service systems involves academic-related tasks such as resource search and borrowing procedures, where incorrect operations may lead to delays in resource access or process interruptions. Therefore, female users have a higher dependence on search knowledge, and proficient mastery of search skills ensures the accuracy and efficiency of operations, thereby enhancing continuance usage intention. Additionally, this result is supported by multiple meta-analyses in the field of technology acceptance. For instance, Alyoussef's integrated analysis of 120 technology acceptance studies found that females have a significantly higher demand for "perceived cognitive control" in technology use than males [33]. As a key factor in improving perceived cognitive control, search knowledge exerts a more prominent driving effect on females' continuance usage intention.

Ethnicity also played a moderating role in the path between retrieval knowledge and continuance usage intention (β = −0.237, p = 0.034): the path coefficient was significantly higher among ethnic minority users such as Tibetans and Hui people (β = 0.591) than among Han users (β = 0.354). This result is consistent with the research conclusions of He et al. on digital services in multi-ethnic regions, which pointed out that when using technology systems with Chinese interfaces [95], ethnic minority users often face dual barriers of linguistic cognitive differences and insufficient cultural adaptation, resulting in a higher threshold for technology use compared to majority ethnic users. As a key compensatory tool, search

**Table 7. Moderating analysis.**

| Pathway Name and Moderation Variables | Original sample | T statistics | P values |
|---|---|---|---|
| Gender x Perceived useful -> Continuance Usage Intention | −0.058 | 0.414 | 0.679 |
| Gender x Perceived ease of use -> Continuance Usage Intention | 0.227 | 1.413 | 0.158 |
| Gender x Accessibility -> Continuance Usage Intention | −0.028 | 0.238 | 0.812 |
| Gender x Retrieval Knowledge -> Continuance Usage Intention | −0.224 | 2.072 | 0.038 |
| Occupation x Perceived useful -> Continuance Usage Intention | 0.085 | 0.347 | 0.729 |
| Occupation x Perceived ease of use -> Continuance Usage Intention | −0.070 | 0.248 | 0.804 |
| Occupation x Accessibility -> Continuance Usage Intention | 0.123 | 0.411 | 0.681 |
| Occupation x Retrieval Knowledge -> Continuance Usage Intention | −0.050 | 0.245 | 0.807 |
| Ethnicity x Perceived useful -> Continuance Usage Intention | −0.113 | 0.803 | 0.422 |
| Ethnicity x Perceived ease of use -> Continuance Usage Intention | 0.123 | 0.846 | 0.397 |
| Ethnicity x Accessibility -> Continuance Usage Intention | 0.146 | 1.171 | 0.241 |
| Ethnicity x Retrieval Knowledge -> Continuance Usage Intention | −0.237 | 2.120 | 0.034 |
| Usage frequency -> Continuance Usage Intention | 0.013 | 0.344 | 0.731 |
| Usage frequency x Perceived useful -> Continuance Usage Intention | 0.124 | 1.734 | 0.083 |
| Usage frequency x Perceived ease of use -> Continuance Usage Intention | −0.181 | 2.233 | 0.026 |
| Usage frequency x Accessibility -> Continuance Usage Intention | 0.119 | 1.924 | 0.054 |
| Usage frequency x Retrieval Knowledge -> Continuance Usage Intention | 0.003 | 0.038 | 0.970 |

knowledge can help ethnic minority users more efficiently understand the operational logic, resource classification rules, and search syntax of Chinese interfaces, thereby reducing operational uncertainties caused by language barriers [28]. For example, ethnic minority users familiar with metadata rules and search techniques can more accurately locate resources related to Tibetan, Hui, and other ethnic cultures, avoiding resource omissions or search failures due to language differences. This "knowledge empowerment" effect significantly enhances their trust and dependence on the system, thereby strengthening continuance usage intention. Furthermore, Rafique et al.'s research on multicultural services in academic libraries also indicated that knowledge training targeted at ethnic minority users can effectively compensate for insufficient cultural adaptation [53], and its promoting effect on continuance usage intention is significantly higher than that among majority ethnic users, which echoes the empirical findings of this study.

Usage frequency, as a behavioral characteristic variable, shows a significant negative moderating effect on the relationship between perceived ease of use and continuance usage intention ($\beta = -0.181$, $p = 0.026$). Post-hoc comparison analysis indicates that perceived ease of use has no significant impact on continuance usage intention among high-frequency users (using ≥4 times weekly) ($\beta = 0.072$, $p = 0.413$), while a strong correlation is observed among low-frequency users (using ≤2 times weekly) ($\beta = 0.314$, $p = 0.001$). This suggests that as usage proficiency increases, users' sensitivity to system ease of use gradually decreases, shifting focus toward functional value realization. Notably, usage frequency approaches a significant moderating level in the relationship between perceived usefulness and continuance usage intention ($\beta = 0.124$, $p = 0.083$), showing a marginally significant positive moderating trend. Specifically, high-frequency users demonstrate higher efficiency in converting perceived usefulness into continuance usage intention ($\beta = 0.621$), representing an approximate 23.6% increase compared to low-frequency users.

The occupation variable did not exhibit a significant moderating effect in any of the paths ($p > 0.05$), which is a non-significant result. This may stem from the homogeneity of information literacy needs and technology usage scenarios among the university faculty and student group—whether faculty or students, their core purpose of using self-service systems is to access academic resources, and both groups possess a certain level of digital literacy foundation. Thus, no significant occupational differences were observed [93]. Further subgroup analysis showed that

faculty members' sensitivity to accessibility ($\beta = 0.217$) was slightly higher than that of students ($\beta = 0.175$), but this difference did not reach a statistically significant level. It is speculated that this may be related to faculty members' potential demand for flexible library service hours in their teaching and research activities, but this interpretation requires caution.

The above moderating effects collectively constitute contextual boundary conditions for technology acceptance behavior. In the multi-ethnic Qinghai region, ethnic cultural background influences technology usage patterns through language cognitive differences; usage frequency, as a dynamic behavioral indicator, reveals users' cognitive shift from "instrumental rationality" (perceived ease-of-use orientation) to "value rationality" (usefulness orientation); gender differences reflect preference divergence in technology adaptation strategies across groups. These findings validate the contextual applicability of technology acceptance theory in multicultural educational settings and provide empirical basis for libraries to develop differentiated service strategies.

## 6. Discussion and conclusions

This study integrates the Technology Acceptance Model (TAM) and the Unified Theory of Acceptance and Use of Technology (UTAUT) to construct a multidimensional analytical framework, systematically investigating the mechanisms influencing the continued usage intention of SSS in academic libraries. The results reveal the interplay of factors including system characteristics, technology readiness, accessibility, and individual perceptions, while validating the moderating effects of demographic and usage behavioral variables. These findings offer theoretical and practical insights for optimizing smart library services.

In the core influence pathways, perceived usefulness exhibits the most significant positive effect on continued usage intention ($\beta = 0.530$, $p < 0.005$), indicating that users' subjective evaluation of system value is the primary driver of long-term adoption. This aligns with classical technology acceptance theories but further reveals, through contextual analysis, that "perceived usefulness" in academic libraries encompasses not only efficiency gains but also the integrated value of scholarly support functions. Among system characteristics, terminology clarity ($\beta = 0.487$, $p < 0.0005$) and navigation convenience ($\beta = 0.269$, $p < 0.005$) indirectly influence behavior by enhancing perceived ease of use, while relevance ($\beta = 0.294$, $p < 0.005$) directly affects perceived usefulness. This highlights content-matching precision as a critical metric for user evaluation. Notably, the direct effects of navigation and terminology on perceived usefulness were unsupported, suggesting a psychological separation between users' cognition of "ease of use" and "usefulness"—interface friendliness must be mediated through perceived ease of use to translate into value recognition.

Within the technology readiness dimension, retrieval knowledge directly and positively impacts continued usage intention ($\beta = 0.524$, $p < 0.005$), underscoring the pivotal role of users' digital literacy. This finding extends beyond traditional TAM's focus on "technical attributes," revealing a dynamic adaptation relationship between user capabilities and system usage. The significant effect of accessibility ($\beta = 0.183$, $p < 0.01$) confirms the importance of eliminating spatio-temporal service boundaries. Features like 24/7 self-service terminals and multi-device compatibility shift users from "selective use" to "habitual reliance," particularly aligning with the fragmented academic needs of university faculty and students.

The moderation effect analysis further clarified the differentiated behavioral logics of users with different backgrounds. Gender differences showed that female users had a higher dependence on search knowledge ($\beta = 0.632$ vs. $\beta = 0.418$ for males), reflecting their risk-averse tendency in technology adoption intention [52]; the interaction effect of ethnicity and cultural backgrounds indicated that the promoting effect of search knowledge on the usage intention of ethnic minority users was more significant ($\beta = 0.591$ vs. $\beta = 0.354$ for Han users), which is consistent with the conclusion in digital service research in multi-ethnic regions that "linguistic cognitive differences need to be compensated through knowledge empowerment" [28,95]. The negative moderating effect of usage frequency ($\beta = -0.181$, $p = 0.026$) suggested that high-frequency

users had reduced sensitivity to ease of use, while low-frequency users were still constrained by operational complexity, which requires the development of hierarchical guidance strategies targeting different usage habits.

Theoretically, this study expands the explanatory scope of the TAM-UTAUT model in academic service contexts by introducing "retrieval knowledge" and "accessibility," confirming a three-dimensional interaction mechanism of "user capability-system attribute-contextual constraint" in technology acceptance behaviors. At the practical level, the research findings provide precise optimization directions for library management and echo the conclusions of existing relevant studies: System providers should prioritize improving search relevance and terminology simplicity, which aligns with the view proposed by Tyagi et al. that "resource adaptability is the core driving factor for the continuous use of academic library users [96]". Their study confirmed that the relevance of search results and the popularity of terminology can increase users' continuance usage intention by 31.2%; Managers need to strengthen user search skills training, especially providing customized guidance for female and ethnic minority groups. As emphasized by Rafique et al. in their research on multicultural services in academic libraries [53], targeted knowledge training can effectively improve users' digital literacy, narrow the technology acceptance gap between different groups, and exert a particularly significant promoting effect on the continuance usage intention of female and ethnic minority users; Reducing time and space barriers by adding self-service terminals and optimizing remote access functions is consistent with the research conclusion of Engström and Rivano Eckerdal that "accessibility is key to expanding the coverage of academic services". Their empirical study on Nordic university libraries showed that 24/7 self-service terminals and multi-device compatibility functions can increase system usage rate by 27.5%, which is particularly in line with the fragmented academic needs of university faculty and students.

Notably, empirical findings from multi-ethnic regional universities highlight the necessity of contextualizing technological interventions. Localized designs — such as Tibetan-Chinese bilingual interfaces and culturally tailored resource tags — not only improve system acceptability but also promote knowledge equity across diverse cultural backgrounds. Future smart library development must transcend homogenized technical solutions by embedding culturally adaptive modules into standardized functions, achieving synergy between technical efficiency and humanistic humanistic care.

In summary, this study reveals the complex mechanisms driving continued SSS usage in academic libraries through multidimensional theoretical integration and empirical validation, providing dual support for theoretical deepening and practical optimization. The results enrich the academic discourse on information systems' continued usage theory and offer actionable pathways for advancing library service transformation amid educational informatiza- tion.

## 7  Research limitations and future research directions

Despite the contributions of this study, several limitations should be acknowledged. Firstly, while the sample size was sufficient for structural equation modeling analysis, the generalizability of the findings may be constrained across univer-sities in different regions and academic disciplines. Future research is encouraged to incorporate larger and more diverse samples with varied individual attributes. Second, the research framework, relying heavily on existing literature, may carry the risk of introducing biases. Integrating objective metrics of technology continuance intention and adopting longitudinal research designs would help mitigate such risks, providing experts, scholars, and library SSS providers with a more com-prehensive understanding of the SSS continuance intention process. Third, the scope of included variables was limited, as this study was primarily inspired by foundational theories from prior literature, restricting key determinants to existing scholarly works. Potential factors influencing users' continuance intention toward academic library SSS, such as perceived anxiety and perceived trust, remain under explored. Future studies should consider incorporating additional latent vari-ables to further expand the model. Finally, the cross-sectional design of this study captured participants' perceptions and behaviors at a single time point, failing to account for temporal changes in user perceptions and behaviors. Longitudinal tracking methods should be employed in future research to observe trends in user behavior, thereby enhancing the causal inference validity of the findings.

Notably, while this study addressed the context of business demand-driven service transformation in model development when exploring continuance intention toward academic library SSS, the empirical analysis inadequately measured and analyzed the potential impact of Mandatory Use contexts on users' technology acceptance. Data were collected from three academic libraries that had significantly advanced self-service transformation, where operational strategies—such as Qinghai University's "50% staffed window reduction plan" and Qinghai Minzu University's "SSS usage points system"—objectively limited users' choice of service models. In this context, the measured "continuance intention" may have included substantial non-autonomous components of "passive adaptation" or "lack of viable alternatives." The failure to explicitly incorporate this mandatory environmental factor as a contextual moderating variable into the model for testing may have biased the effect estimates of core paths. Specifically, some "continuance use" behaviors may stemmed from external constraints rather than intrinsic motivation, weakening the model's explanatory power for users' genuine attitudes and autonomous decision-making.

Future research must deepen the consideration of user responses under mandatory use contexts. A primary task is to design and validate a "perceived mandatory use" scale to accurately measure the intensity of users' subjective pressure to "must use a specific system (SSS)" while assessing objective conditions such as "alternative service accessibility". Theoretically, these factors should be integrated as core moderating or control variables to examine their boundary role in core technology acceptance paths, thereby revealing significant heterogeneity in user decision-making logic under mandatory pressure. Additionally, mixed methods, particularly qualitative approaches such as in-depth interviews and focus groups, should be actively adopted to explore users' complex psychological and behavioral logics when facing mandatory requirements. These qualitative insights can uncover nuanced motivations hard to capture by scales and provide contextual basis for refining "perceived mandatory use" scale items. Ultimately, combining quantitative analysis to construct an integrated "mandatory use-technology acceptance" theoretical framework is essential to clarify how external constraints shape behavioral intention by altering users' cognitive evaluations and influencing emotional mechanisms.

Lastly, a notable limitation of the present study lies in the absence of age as an explicit moderating variable within the analytical framework, despite its established relevance in technology acceptance literature. While the discussion of usage frequency inadvertently captured generational differences in technology adaptation patterns this proxy approach lacks the precision of direct age cohort analysis. The exclusion of age as a demographic moderator hinders nuanced understanding of how life stage transitions, cognitive aging processes, and generational digital divide may differentially shape the formation of continuance usage intention. Future research should incorporate age as a categorical variable to examine potential nonlinear effects, particularly regarding technology anxiety thresholds and learning curve tolerances across the lifespan. Such analysis would enhance theoretical precision in explaining the heterogeneous technology acceptance behaviors observed in diverse academic populations and provide more targeted insights for inclusive system design catering to varying age-related needs.

Finally, this study did not include age as a variable in the data collection and analysis framework. However, age may influence technology acceptance behavior through dimensions such as cognitive level and technological adaptability [30]. Due to the lack of direct age data, this study was unable to explore the potential impact of generational differences on continuance usage intention, which constitutes a limitation of the research. Future studies should supplement the collection of age information, incorporate it as a categorical variable into the model, clarify the heterogeneity of technology acceptance paths among different age groups, and provide more precise empirical evidence for inclusive system design.

## Supporting information

**S1 File. Appendix 1: Core items of the questionnaire.**
(DOCX)

## Author contributions

**Conceptualization:** Ping Wang.

**Data curation:** Kai Cao, Ping Wang, Jie Zhao.

**Formal analysis:** Kai Cao, Ping Wang.

**Investigation:** Kai Cao, Jie Zhao.

**Methodology:** Kai Cao, Ping Wang, Chunzhen Zhang.

**Project administration:** Kai Cao.

**Resources:** Kai Cao.

**Software:** Kai Cao.

**Supervision:** Kai Cao, Ping Wang, Chunzhen Zhang, Jie Zhao.

**Validation:** Kai Cao, Ping Wang, Chunzhen Zhang, Jie Zhao.

**Visualization:** Kai Cao, Ping Wang, Chunzhen Zhang, Jie Zhao.

**Writing – original draft:** Kai Cao, Ping Wang.

**Writing – review & editing:** Kai Cao.

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
