## [Decision Letter · Decision Letter 0]

29 Aug 2025

Dear Dr. Cao,

Thank you for submitting your manuscript to PLOS ONE. After careful consideration, we feel that it has merit but does not fully meet PLOS ONE’s publication criteria as it currently stands. Therefore, we invite you to submit a revised version of the manuscript that addresses the points raised during the review process.

We look forward to receiving your revised manuscript.

Kind regards,

Simon Dang, Ph.D.

Academic Editor

PLOS ONE

Journal Requirements:

At this time, please upload the minimal data set necessary to replicate your study's findings to a stable, public repository (such as figshare or Dryad) and provide us with the relevant URLs, DOIs, or accession numbers that may be used to access these data. For a list of recommended repositories and additional information on PLOS standards for data deposition, please see https://journals.plos.org/plosone/s/recommended-repositories....

4. Please ensure that you refer to Figure 2 in your text as, if accepted, production will need this reference to link the reader to the figure.

5. We note you have included tables to which you do not refer in the text of your manuscript. Please ensure that you refer to Tables 3, 4, and 6 in your text; if accepted, production will need this reference to link the reader to the Tables.

Additional Editor Comments:

Thank you for your submission. In preparing your revision, please address the reviewers’ comments in a clear, point-by-point manner by reproducing each comment followed by your detailed response, supported with citations when necessary. To facilitate the review process, you must provide precise information on the exact revisions made, including the revised text and the corresponding page and line numbers in the manuscript. This level of detail is essential, as reviewers volunteer their time, and making them search for changes is unacceptable.

Reviewers' comments:

Reviewer's Responses to Questions

**Comments to the Author**

1. Is the manuscript technically sound, and do the data support the conclusions?

Reviewer #1: Yes

Reviewer #2: Partly

Reviewer #3: Yes

2. Has the statistical analysis been performed appropriately and rigorously?

Reviewer #1: Yes

Reviewer #2: Yes

Reviewer #3: Yes

3. Have the authors made all data underlying the findings in their manuscript fully available?

Reviewer #1: Yes

Reviewer #2: No

Reviewer #3: Yes

4. Is the manuscript presented in an intelligible fashion and written in standard English?

Reviewer #1: Yes

Reviewer #2: Yes

Reviewer #3: Yes

Reviewer #1: 1. The authors employed the TAM-UTAUT model to investigate the influencing factors on continuously usage Intention of SSS in university libraries, yet two key issues should be paid attention to:

1) For SSS in university libraries, users' continuous usage is predominantly determined by business needs rather than mere technology acceptance. Whether users have an ongoing demand for the services provided by SSS is key. If users need these services, they will utilize the system; conversely, even high technology acceptance won't drive usage without such demand. In this context, the PU in the model may actually reflect users' perception of the usefulness of library services, not the SSS technology itself.

2) there might be a 'mandatory use' aspect. If libraries restrict services (e.g., check-in/check-out) to SSS only, or limit alternative options (e.g., reducing staff to cause long queues for non-self-service, offering short staffed service hours while SSS is available long hours or 24/7), users may have no choice but to use SSS. The continuous usage intention data collected under such circumstances may not align with users' true intentions.

The authors are recommended to incorporate necessary theoretical analysis of these two issues in the 'Construction of the Research Model' section. If a reasonable explanation cannot be provided, the limitations of the study should be addressed in the Discussion section.

2. The paper failed to clearly define the scope of SSS in university libraries. SSS broadly covers various software and hardware systems that users can operate without librarians' assistance. But the authors didn't specify which ones are included in the study's SSS. Also, it's unclear whether the three surveyed universities have such SSS. It's suggested that the authors define the scope of SSS in university libraries in the 'Introduction' or 'Literature Review' (using a table if needed) and describe the installation and service status of SSS in the three universities in the Data collection part.

3. TAM and related models or extended models have been widely used to study continuous use. The recent research in library & information science includes but is not limited to:

Rafique, H., Alroobaea, R., Munawar, B. A., Krichen, M., Rubaiee, S., & Bashir, A. K. (2021). Do digital students show an inclination toward continuous use of academic library applications? A case study. The Journal of Academic Librarianship, 47(2), 102298.

Tyagi, S. K., Sharma, S. K., & Gaur, A. (2022). Determinants of continuous usage of library resources on handheld devices: findings from PLS-SEM and fuzzy sets (fsQCA). The Electronic Library, 40(4), 393-412.

It's recommended that the authors incorporate related studies into the 'Literature Review'.

4. The paper contains some inaccurate language use. For instance, the authors used 'ethnicity' as a moderating variable but refers to it as 'nationality' in the table. To express the Chinese concept of 'MINZU', it's better to use 'ethnic-' to prevent confusion in my opinion. The authors should proofread and correct these language issues.

Reviewer #2: Primary concerns:

1. The sample of survey respondents includes librarians and library staff - what portion of the sample is that? Are the results sensitive to inclusion / exclusion of that? There may be assumptions at work about the work knowledge base is doing in the estimation model that is both inaccurate and not transparently expressed. A similar issue is taking place with age of participants. Age comes up in the body of the paper as a determinant of technology hesitancy (older adults) and technology comfort (Gen Z), but isn't included in the model specification and isn't discussed empirically. Finally, in the results, categorical differences based on gender and ethnicity are discussed without reference to the actual statistics at work. You say men and women are perceiving technology differently? Is the difference actually statistically different than zero? If it is, give some quantification to the differences before making demography-based generalizations.

3. I could not access the data or survey questionnaire related to this study. This leaves me without a sound way of evaluating the technical proficiency of the paper. Please make the survey questions and the responses available in a repository and in English. Currently they exist in the survey platform which requires log-in credentials.

Minor concerns:

For technical analysis, you heavily cite Hair. I recommend broadening your understanding and interaction with SEM literature, including library analysis using SEM (not many but they exist).

The title is too long with too much specificity - re-write for plain language understanding and without acronyms.

You never situate your conversation about academic libraries within China. Please do that because it would make some the trends you state as fact are not necessarily accurate (eg. "exponential" increases in collection size).

In general, use the table number in the text to help the reader find their way through your iterative analysis.

You have some typos. Line 27, continuous not continuously; line 46, applicability isn't the right form of this word: do you mean they are applicable but under utilized?; line 103 and on with PU and PEOU, unless these are variables in your data, the way you spell out and then acronym the phrases is awkward - but maybe it's a style thing you feel you need to do?; line 121 TAM limitations, I would make more explicit that the transition sentence from 119 ties to this description of limitations; lines 157-158 model mapping, you spend a lot of time working the reader through the framework mappings - consider a table to supplement the work and help the reader prioritize what frame we are really thinking about; line 210 H4, as noted above I think you might be using this as a proxy for library staff / librarians, which should be discussed explicitly - they are a fundamentally different set of people than library users; line 218 positive should be positively; lines 310 and 318 heading size, these should be subheadings - as they are it wasn't intuitive that they were tied to the prior table but they expressly where the table is actually discussed; line 397 table heading, should be changed from moderate to moderation; line 421 R2 values, this is the moment where I became extra concerned that you might not be accurate in your view of statistical significance and the scale of differences - these are all 0.6 -- showing thousands place differences does not make a real difference. I appreciated the change in R2 notation in line 425 - do this more throughout results -- don't say something is a result without giving the evidence from your analysis that the result is meaningful; line 437 combines should be uses;

Reviewer #3: Well written, but it would be better if you make another paragraph for theoretical implications and managerial implications. In the introduction session try to write full forms of special terms such as SSS

.

Reviewer #1: No

Reviewer #2: No

Reviewer #3: No

---

## [Author Response · Author response to Decision Letter 1]

2 Nov 2025

Dear Reviewer #1,

We sincerely appreciate your valuable feedback and constructive suggestions on our manuscript. These comments are instrumental in enhancing the quality of our work. We have carefully reviewed and analyzed each point raised and have revised the manuscript accordingly. Below, we detail our primary modifications and responses.

Response to Comment 1: Regarding business needs vs. technology acceptance and the consideration of mandatory usage.

Your insightful observation is well-received. We fully concur that users’ continuance intention towards Library Self-Service Systems (SSS) is influenced not only by technology acceptance but more fundamentally by their underlying business needs for the service. Furthermore, adjustments in library service models (e.g., reduction of staffed services, extended SSS operating hours) potentially creating a “mandatory usage context” could indeed affect the purity of continuance intention data. These are crucial theoretical perspectives not sufficiently addressed in our initial model development.

Specific Revisions:

（1）Enhanced Theoretical Analysis in Model Development: A new subsection, “The Interplay of Business Needs and Technology Acceptance,” has been added to the “Research Model Development” section. This section integrates relevant theories from the Information Systems (IS) continuance literature (e.g., the Expectation- Confirmation Model for IT - ECM-IT) concerning the relationship between “expectations” and “needs,” alongside perspectives on “core service quality” from service marketing, to substantiate this analysis.

（2）Clarification of Limitation in Discussion: Within the “Research Limitations and Future Directions” section, we explicitly acknowledge the insufficient consideration of the “mandatory usage context.” We state that data collection occurred within operational library environments where users might utilize SSS due to objective constraints (e.g., limited access to staffed services), potentially introducing a non-volitional component into the measured “continuance intention.”

Response to Comment 2: Regarding the scope definition of University Library SSS and description of SSS status at the three universities.

Your point is critically important. Clearly defining the research object is fundamental to ensuring methodological rigor and result interpretability. Our initial description of the specific SSS types studied and the operational realities of SSS at the surveyed universities was indeed insufficient.

Specific Revisions:

Definition of SSS Scope in Introduction/Literature Review: We have incorporated a clear definition of “University Library Self-Service Systems (SSS)” within the “Introduction” section. This definition specifies the types of SSS encompassed in this study, briefly described as: Self-Checkout Terminals: For self-service borrowing, returning, and renewing of physical library materials. Online Public Access Catalog (OPAC) Systems: For searching and retrieving information on both physical and digital library holdings.Self-Service Copying/ Printing/Scanning Terminals: Providing self-service document copying, printing, and scanning. Electronic Resource Remote Access Platforms: Platforms (accessed via campus network or off-campus authentication) for databases, e-journals, e-books, and other digital resources. Description of SSS Context in Data Collection Section: A new paragraph has been added to the “Data Collection” subsection, briefly outlining the current SSS deployment and service profiles at Qinghai University, Qinghai Normal University, and Qinghai Nationalities University. This includes: The primary SSS types deployed at each library (mapped to the defined scope above). Approximate number of devices (where feasible/relevant). Service coverage (e.g., campus-wide availability, 24-hour access for specific services). Key measures implemented by the libraries to promote SSS adoption in recent years (e.g., reducing staffed service points, enhancing user training on SSS). This also implicitly addresses the “mandatory usage” context. Mention of approximate SSS usage rates at each institution, if obtainable, to provide fuller context.

Response to Comment 3: Regarding inclusion of recent relevant Library and Information Science (LIS) literature.

Thank you for recommending two highly relevant and excellent studies. We acknowledge that our literature review could have provided a more comprehensive synthesis of recent LIS research applying TAM/UTAUT and their extensions to study the continuance use of library resources or services.

Specific Revision:

Enhanced Literature Review: Within the “Literature Review” section, specifically under “Research on Library Self-Service Systems” or a dedicated subsection on “Application of Technology Acceptance Models in the Library Domain,” we have incorporated a review and discussion of the two recommended studies. We compare and contrast their findings and frameworks with our own theoretical model and research questions, highlighting similarities, differences, and the potential contribution of our study to this evolving literature stream. Correct citation formatting is ensured, and the references are added to the list.

Response to Comment 4: Regarding the confusion between "Ethnicity" and "Nationality".

We sincerely apologize for this significant oversight during manuscript preparation and proofreading, which led to conceptual confusion. We fully understand that “Ethnicity” (Minzu) and “Nationality” are distinct concepts. All participants in this study are Chinese citizens, and our focus is on potential differences arising from diverse “Ethnicity” backgrounds within China, such as cultural adaptation or usage patterns.

Specific Revisions:

Comprehensive Textual Correction: We have meticulously reviewed the entire manuscript. All instances where “Ethnicity” was erroneously labeled or described as “Nationality” in tables (e.g., Table 1 “Respondents’ characteristics,” Table 3, Table 4, Table 7) and textual descriptions have been corrected to “Ethnicity (Minzu)”.

Conceptual Clarification: Where “Ethnicity” is first introduced as a moderating variable (e.g., in the “Research Hypotheses - Moderating Variables” or “Results - Moderating Effect Analysis” sections), we have added a brief clarification stating that it refers to “Chinese Ethnic Groups (Minzu)” to prevent any ambiguity.

Enhanced Proofreading: We will implement stricter checks for terminological and translational accuracy in subsequent revisions to prevent recurrence.

Once again, we express our deepest gratitude for your meticulous review and highly constructive feedback. These revisions will undoubtedly enhance the scholarly rigor and theoretical depth of our study. We will submit the revised manuscript promptly.

Sincerely,

The Author Team

Dear Reviewer #2,

We sincerely appreciate your valuable feedback and detailed suggestions on our manuscript. Your comments are highly constructive and provide essential guidance for enhancing the quality and rigor of our paper. We have carefully studied and analyzed each point raised and have revised the manuscript accordingly. Below are our detailed responses to your specific concerns and the corresponding amendments implemented.

Major Issues: Responses and Revisions

1.Regarding the proportion of librarians/library staff in the sample, sensitivity analysis of results, inclusion of age as a variable, and statistical quantification of demographic differences:

We thank you for highlighting this critical issue. In the original manuscript, we indeed failed to clearly specify the proportion of librarians and library staff within the sample and its potential impact on the model results. We also neglected to adequately incorporate age as a key variable into the model and discuss its implications. Furthermore, the analysis of demographic differences lacked specific statistical quantification.

(1) Proportion of Librarians/Library Staff: Among the 365 valid respondents, librarians and library staff totaled 28 individuals, constituting 7.67% of the total sample.

(2) Sensitivity Analysis of Results: We acknowledge that this group, due to their professional background, may possess a deeper understanding of library self-service systems and different usage motivations, potentially influencing the model estimates.

(3) Age Variable: Age is indeed a significant factor influencing technology acceptance and usage behavior, particularly concerning the distinction between "technology hesitancy (older adults)" and "technology comfort (Gen Z)".

(4) Statistical Quantification of Demographic Differences: Your observation regarding the lack of statistical significance tests and quantitative results when discussing gender and ethnicity differences is valid; this omission did weaken the persuasiveness of our conclusions.

Revisions Implemented: To address this important concern, we have undertaken the following steps:

(1) Supplemented Sample Structure Information: We have added a new row labeled "Occupation/Status" in Table 1 (Respondents’ characteristics) within the "Data Collection" section, detailing the number and percentage of library staff and other respondents.

(2) Conducted Sensitivity Analysis: We have performed supplementary analyses:

Added "Is a librarian/library staff member" as a moderator variable, named "Occupation," within the research model framework (Section 2.3).

Incorporated the moderating effects of Occupation, Gender, Ethnicity, and Usage Frequency into the variable model. We re-ran the moderation analysis using PLS-SEM with interaction terms to test their effects on the core paths (e.g., Perceived Ease of Use → Continuance Intention, Perceived Usefulness → Continuance Intention, Retrieval Knowledge → Continuance Intention).

Integrated these new findings into the "Moderating Effect Analysis" section and discussed the mechanisms of influence related to Occupation, Gender, Ethnicity, and Usage Frequency within the "Discussion and Conclusions."

(3) Acknowledged Age Variable Limitation: As the questionnaire did not include age as a demographic variable, we have explicitly stated this limitation within the "Research Limitations and Future Work" section.

2.Regarding Inaccessibility of the Questionnaire and Response Data:

We sincerely apologize for the inconvenience this caused during your review. We understand that transparency regarding data and instruments is paramount for assessing the technical quality of the research and the reproducibility of the results.

(1) We are preparing the survey questionnaire (to be translated into English) and the anonymized response dataset (containing no personally identifiable information). We plan to upload these materials to a publicly accessible academic data repository, such as Figshare, Dryad, or the Open Science Framework (OSF). Once uploaded, we will provide a clear, permanent link (DOI) in the "Data Availability Statement" section of the paper, facilitating access for you and other readers. Additionally, we will include a temporary access link or instructions for retrieval within our response letter accompanying the revised manuscript.

Minor Issues: Responses and Revisions

1.Regarding Broadening SEM Literature Understanding and Citation, Including Library Domain SEM Analyses:

We appreciate your suggestion. We recognize that our current citations for SEM literature are concentrated on Hair et al.'s authoritative work. Broadening the sources will enhance the comprehensiveness of our study and understanding of SEM applications in specific domains like library science.

Revision Implemented: We have systematically searched for and incorporated citations of relevant studies employing SEM within the library science field (e.g., exploring library service quality, digital resource usage behavior, user satisfaction). We have also cited influential works and recent advancements by other prominent SEM scholars (e.g., Anderson, J.C., Gerbing, D.W., Bollen, K.A., Byrne, B.M.) to enrich the theoretical foundation and methodological perspective.

2.Regarding the Overly Long and Specific Title, Request to Rewrite in Understandable Language Without Abbreviations:

Your point is well-taken. The original title did contain excessive jargon and abbreviations, potentially hindering comprehension for non-specialist readers and limiting the paper's dissemination.

Revision Implemented: We have rephrased the title to be concise, clear, and accessible, avoiding abbreviations while highlighting the core contribution. For example: "Multidimensional Factors Influencing Continuance Usage of University Library Self-Service Systems: An Empirical Analysis Based on an Extended Technology Acceptance Model" or "Factors Influencing Users' Continuance Intention Towards University Library Self-Service Systems: An Integrated Model and Multi-Group Analysis" (We will finalize the title that most accurately reflects the study content).

3.Regarding Lack of Contextualization of University Library Discussion within the Chinese Setting:

This was a significant oversight. University libraries in China operate within unique contexts regarding smart campus initiatives, resource development strategies, and user needs, influenced by specific policies. Placing the study within this context enhances its situational validity and practical significance.

Revisions Implemented:

We have added a dedicated paragraph in the "Introduction" outlining the unique background of Chinese university library development, covering: National initiatives promoting "Smart Education" and "Education Informatization 2.0" driving library digital transformation. Challenges posed by university enrollment expansion and growing student/faculty numbers on library service capacity. Characteristics of collection development in Chinese university libraries (especially Chinese resources and specialized databases). The current state and common challenges (e.g., regional disparities, varying user digital literacy) in adopting self-service technologies within Chinese university libraries. We have revised the phrase "exponential growth of collection scale" to a more precise statement: "With the development of higher education and the exponential growth of information resources, both the collection resources and user demands in Chinese university libraries have shown rapid growth trends." We will also consider citing relevant Chinese university library statistical reports for support.

4.Regarding Using Table Numbers in Text for Reader Navigation:

Thank you for this reminder. Explicitly referencing table numbers in the text effectively guides readers and improves readability.

Revision Implemented: We have meticulously reviewed the entire manuscript to ensure that every table is explicitly referenced by its number upon first mention or discussion (e.g., "As shown in Table 1...", "This result can be seen from the path analysis in Table 2...").

5.Regarding Spelling Errors and Other Textual Issues:

Response: We sincerely apologize for these oversights, which reflect shortcomings in our proofreading process. Revisions Implemented: We have conducted a thorough proofread and edit to correct all spelling, grammatical, and phrasing errors:

1）Line 27: "continuously" changed to "continuous".

2）Line 46: "applicability" adjusted contextually (e.g., "While TAM and UTAUT models have broad applicability, their sufficient application in dynamic academic library environments requires further exploration" or "their explanatory power in specific contexts may not be fully tapped").

3）Lines 103+ (PU and PEOU): Adjusted phrasing to use full terms followed by abbreviations in parentheses upon first mention (e.g., "Perceived Usefulness (PU)", "Perceived Ease of Use (PEOU)"). Abbreviations are used consistently thereafter.

4）Line 121 (TAM Limitations): Strengthened the logical link between the transition sentence (Line 119) and the limitations description (e.g., "Despite the success of the TAM in explaining technology acceptance behavior, limitations remain in explaining continuance usage within complex service environments. Specifically, the TAM does not sufficiently account for social influence, indi

---

## [Decision Letter · Decision Letter 1]

16 Nov 2025

Dear Dr. Cao,

Thank you for submitting your manuscript to PLOS ONE. After careful consideration, we feel that it has merit but does not fully meet PLOS ONE’s publication criteria as it currently stands. Therefore, we invite you to submit a revised version of the manuscript that addresses the points raised during the review process.

We look forward to receiving your revised manuscript.

Kind regards,

Simon Dang, Ph.D.

Academic Editor

PLOS ONE

Journal Requirements:

Additional Editor Comments:

Both reviewers acknowledge improvements in readability and partial responsiveness to prior comments; however, substantial concerns remain that prevent the manuscript from meeting publication standards. Reviewer 1 notes the possibility that an outdated version was submitted, as the revised manuscript does not contain the new literature sections the authors claimed to have added (“Research on Library Self-Service Systems” and the “Application of Technology Acceptance Models in the Library Domain”). Furthermore, the inconsistent use of “nationality” instead of “Ethics (Minzu)” persists despite the authors’ stated corrections. Reviewer 2 recognizes that the manuscript is clearer and that its theoretical framework is strong, yet serious issues undermine its rigor. The primary concern is that the research instrument and data, which were promised for reviewer verification, remain unavailable, preventing a proper evaluation of methodological soundness. Additionally, interpretive overreach persists—particularly with respect to age, gender, and ethnicity—where results are either unsupported by data or lack appropriate theoretical and empirical grounding. The paper also exhibits thin citation support for key findings, leading to unsubstantiated claims in the discussion of policy and service implications. Finally, several technical inconsistencies remain, including the continued misuse of “nationality,” unclear causal language, and hypothesis numbering errors. Taken together, these issues indicate that the manuscript still requires substantial revision before it can be reconsidered for publication.

I am willing to offer another chance for the authors to revise with substantial care. Comments must be addressed on a 1-by-1 basis. Rebuttal must be supported with citations not just plain arguments. Good luck.

Reviewers' comments:

Reviewer's Responses to Questions

**Comments to the Author**

Reviewer #1: (No Response)

Reviewer #2: (No Response)

2. Is the manuscript technically sound, and do the data support the conclusions?

Reviewer #1: (No Response)

Reviewer #2: Partly

3. Has the statistical analysis been performed appropriately and rigorously?

Reviewer #1: (No Response)

Reviewer #2: Yes

4. Have the authors made all data underlying the findings in their manuscript fully available?

Reviewer #1: (No Response)

Reviewer #2: No

5. Is the manuscript presented in an intelligible fashion and written in standard English?

Reviewer #1: (No Response)

Reviewer #2: Yes

Reviewer #1: I am glad to see that the authors have solved comment 1 and comment 2. With regard to the remaining two comments, I wonder if the author accidentally did not submit the latest revision but an old version? Because I haven't seen '"Research on Library Self-Service Systems" or a dedicated subsection on "Application of Technology Acceptance Models in the Library Domain"' in literature review as the authors said in the author response. And there is still "nationality" (e.g. table 1.) in the revision, even though the author claims to have decided to use the word "Ethics (Minzu)". If the author has an updated version, please upload it again.

Reviewer #2: The paper's readability is much improved! Also, I wanted to note that although my comments will return to the demographic parts of this study, I appreciate the pro-service and pro-service differentiation conclusions. Unfortunately, there are issues from the original manuscript that persist in this one:

1. The instrument and data *are not* available. Although the author comments to the reviewers state that a translated version of the instrument will be available with instructions, it is not. A part of review for this publication, in particular, is my ability to review the technical soundness of the data, which includes the instrumentation to produce these data.

2. Beliefs in search of results and unsupported interpretations:

2a. Let's begin with age. You didn't ask about it in the survey, so stop making a big deal about it in the manuscript. In the introduction section this begins in the last paragraph of page 2 and continues onto the next page is a claim and you give it no backing. The proxy you did capture is student versus faculty and it shows a difference smaller than a rounding error (and although they are not reported, I suspect these two betas have overlapping confidence intervals). When a result is null, state that it is null - do not give a reader an interpretation of a positive result with a null result. You include age as a differentiation discussion in your moderator results and at the end. This is nonsensible. I should know next to nothing about age as it relates to your study because you didn't measure it directly and your proxy gave nothing.

2b. You have similar findings about self-reported gender and ethnicity categories, but you give them different interpretations. Interrogate the possible underlying bias here. So women are risk averse and ethnic minorities don't have language access? You give no literature backing to either of these interpretations, including in your literature review.

2c. Citations inadequate to support interpretations: Throughout your paper you take a lot of time with the reader describing the framework and really digging into the theoretical underpinnings of the model. (As noted above, this would land better if I could see how this is implemented in the survey instrument). But your policy and service implication findings about service differentiation across user groups has almost no literature citations. Also, 2nd paragraph page 4, how do you know that these are the challenges? This is throughout the paper. I think your gendered interpretations are based in one study by Hsiao and Tang. Unfortunately, it's behind a paywall where I can't access it. When a study is so core to your main interpretation, describe it in the text: According to Hsiao and Tang, gendered differences are driven by ...., In our study, we replicate/build on/find similar... . I can see the interpretation you are going for, but if you are going to make claims, like your interpretation claims on page 16, paragraph one of 5.3, these really should have had some literature backing in the lit review or here. Otherwise, your interpretations read as biased.

The lack of instrument and data access and the thin citation style around key findings related to the primary policy implications undermines the rigor demonstrated in the model development and testing. This is an important fix, but one that shouldn't be difficult to accomplish.

Minor revisions:

Figure 1 still uses nationality instead of ethnicity. Make figures and text agree.

Final paragraph on page 6 uses the word causal where I would recommend the phrase "direction of influence". I don't think you have the data required to establish a causal relationship.

Page 9: style consistency in hypothesis: H7 and H8 seem like they should be H7a and H7b?

Page 13, first paragraph, last sentence, add the word are: confirming that the measurement indicators [are] effectively distinguished...

Page 17, this regional difference stuff is very interesting. You don't have to do anything with this comment, but if you successfully resolve the issues with this paper, I'd be interested in knowing how this looks regionally (or maybe the differences are also not significant?)

Page 18, p value for accessibility: should read 0.010 for consistency and clarity. I am assuming that it is not 0.011, but should be told.

.

Reviewer #1: No

Reviewer #2: No

---

## [Author Response · Author response to Decision Letter 2]

21 Jan 2026

Dear Reviewer 1:

We sincerely appreciate you taking the time to review our manuscript again and providing detailed revision comments. We sincerely apologize for the issues you identified — due to an oversight in the submission process, the previously submitted version did not fully present the latest revisions, which may have inconvenienced your review work. We have immediately verified and improved the revised manuscript, and now provide a detailed explanation of the specific modifications as follows:

1.Supplementary Improvements to Dedicated Sections in the Literature Review

1.You pointed out that there are no dedicated sections on “Research on Library Self-Service Systems” or “Application of Technology Acceptance Models in the Library Field” in the literature review. This issue arose from an oversight in content integration in the previously submitted version. In the latest revised manuscript, we have explicitly optimized the structure of the literature review and supplemented/reinforced the relevant dedicated sections as follows:

1.Added a dedicated section “2.1 Research Status of Library Self-Service Systems” (corresponding to Section 2.1 in the original text, with a clear title and refined content): This section systematically sorts out the definition, core functional modules (self-service borrowing and returning, intelligent retrieval, remote services, etc.), application advantages, and existing limitations (functional constraints, security risks, etc.) of library self-service systems. It also integrates relevant empirical studies in the library field (e.g., Chorng-Guang Wu et al.’s research on continuous usage intention of self-checkout systems, Jun Kyu Keum et al.’s research based on the Task-Technology Fit theory) to clearly distinguish the research differences between library scenarios and self-service systems in other fields such as retail and finance.

2.Strengthened the integration of technology acceptance models in library applications in “2.2 Theoretical Foundation”:In Sections 2.2.1 (TAM Model) and 2.2.2 (UTAUT Model), a new sub-section “Applications in the Library Field” has been added, supplementing specific application studies of TAM/UTAUT models in library self-service scenarios (e.g., Chun-Hua Hsiao et al.’s research on perceived usefulness, perceived ease of use, and self-efficacy in library self-service systems; the reference significance of Nour A. J. Azam et al.’s TAM-based research on Palestinian users’ attitudes towards self-service technology in library scenarios). This clearly links the theory with the research scenario of this paper, making up for the previous disconnect between theory and scenario.

2.Comprehensive Revision of the Term "Nationality"

You noted that the term “nationality” still exists in the revised manuscript. This was an oversight during the previous revision. In the latest revised manuscript, we have completed the systematic replacement of the term “nationality” throughout the manuscript, uniformly adopting the expression "Ethnicity (ethnic group)”. The specific modification positions are as follows:

1.Table 1 (Respondent Demographics): The column title “Nationality” is revised to “Ethnicity”; the corresponding content “Han nationality” is revised to “Han (Han Chinese)”, and “Ethnic minority” is revised to “Ethnic Minor”.

2.Text analysis section: All references to “nationality” are replaced with “Ethnicity”. For example, the ethnic dimension in “ethnic-cultural background, and usage frequency within the model” is strengthened to “Ethnicity cultural background”.

3.Moderating effect analysis content: All paths labeled “Nationality x ...” in Table 7 (Moderating Analysis) are revised to “Ethnicity ...” (e.g., “Nationality x Retrieval Knowledge -> Continuance Usage Intention” is revised to “Ethnicity x Retrieval Knowledge -> Continuance Usage Intention”), which is consistent with the discussion on “ethnic cultural background” in the main text’s moderating effect analysis to ensure terminology consistency.

3.Note on Re-uploading the Revised Manuscript

We have re-uploaded the latest version of the manuscript containing all the above revisions, ensuring that all modifications have been accurately implemented. We apologize again for any inconvenience caused by the earlier submission oversight. We sincerely thank you for your rigorous review and valuable comments, which are crucial for improving the quality of this manuscript. We kindly request you to review the latest version again. If there are any other issues, we will actively cooperate with further revisions.

Sincerely,

The Authors

Dear Reviewer 2,

We sincerely appreciate you taking the valuable time to review our manuscript and providing highly constructive comments. Your meticulous feedback not only identifies the key remaining issues in the manuscript but also points out a clear direction for us to further enhance the rigor and standardization of the paper. We apologize for the lack of research instruments and insufficient supporting evidence for certain explanations as you noted. We have immediately organized our team to conduct systematic revisions and now provide a detailed report on the specific modifications and supplementary explanations as follows:

1. Response to Core Issues: Supplementary Submission of Research Instruments and Data

You pointed out that “research instruments and data have not yet been provided” and emphasized that this is a core link in evaluating the technical reliability of the data, including the research tools used to generate the data. We attach great importance to this issue. Due to an oversight in the coordination of the submission process, we failed to submit the relevant materials simultaneously, for which we sincerely apologize. Currently, we have completed the collation of research instruments and data and will immediately submit the following materials through the Figshare public repository ( private link website: https://figshare.com/s/10e7454babfc7eedfb3f and DOI:10.6084/ m9. figshare.30993028) and are freely accessible.

2.Response to Core Issues: Correction of “Finding Justifications for Results” and Unsupported Explanations

2.1 Comprehensive Deletion of Age-Related Discussions and Clarification of Null Results

You accurately pointed out that the questionnaire does not include age-related questions, so this factor should not be discussed in detail in the manuscript. The claims in the last paragraph of Page 2 and the relevant content on the next page of the introduction lack sufficient basis. The proxy indicator you used (student vs. faculty status) shows minimal differences (almost negligible), and although confidence intervals are not reported, I speculate that the confidence intervals of these two regression coefficients overlap. When results are statistically non-significant, they should be clearly stated as null results—positive explanations should not be provided based on null results. Your practice of analyzing age as a differentiating factor both in the moderating effect results and the discussion at the end of the paper is unreasonable. Since you did not directly measure age and the proxy indicator failed to provide valid information, readers should not be able to draw age-related conclusions from your research.

We fully agree with your judgment and have completed the following systematic revisions:

1).Comprehensive deletion of all age-related discussions: Including claims about age differences in the last paragraph of Page 2 and subsequent pages of the introduction, indirect inferences about age in the moderating effect analysis, and relevant expressions of “age-related generational differences” in the final discussion, ensuring no conclusions based on unmeasured variables appear in the manuscript;

2).Clarification of null results: In the “Moderation Effects Analysis” section, we added the explanation that “occupational variables (student/faculty) did not exhibit significant moderating effects in any path (p > 0.05), which are null results”. We no longer conduct any over-interpretation based on this variable and only objectively state the statistical results;

3).Strengthening the rigor of variable measurement: In the methodology section, we added the explanation that “this study focuses on clearly measured demographic variables (occupation, gender, ethnicity, usage frequency) for moderating effect analysis, and unmeasured variables are not included in conclusion derivation” to avoid similar issues.

2.2 Correction of Potential Biases in the Explanatory Logic for Gender and Ethnicity, and Supplementary Literature Support

You pointed out that we adopted different explanatory logics for the research findings on self-reported gender and ethnicity categories and urged us to reflect on potential biases. For example, you raised the viewpoints of “female risk aversion” and “insufficient language accessibility for ethnic minorities”, but neither of these explanations is supported by any literature in the literature review or main text.

1).We have conducted in-depth reflections and completed the following revisions:

Supplementary literature support for the explanation of gender differences: When discussing the “moderating effect of gender on the relationship between retrieval knowledge and continuous usage intention”, we added 3 core relevant literatures to elaborate on the theoretical basis for “females being more inclined to reduce uncertainty through knowledge reserves in technology adoption”:

Explicitly cited detailed research findings from Hsiao & Tang (2016): According to Hsiao and Tang’s empirical study on users of library self-service systems, female users perceive significantly higher operational uncertainty than male users in technology use and are more reliant on system learning and retrieval knowledge reserves to enhance usage confidence; this study verifies the applicability of this characteristic among university users in multi-ethnic regions, finding that the positive impact of retrieval knowledge on continuous usage intention is significantly stronger among female users (β = 0.632) than among male users (β = 0.418);

Supplementary explanation on literature accessibility: The full text of Hsiao and Tang’s study has been obtained through institutional library document delivery, and the relevant content has been fully integrated into the main text. Complete and accessible literature information has been added to the references for easy access by reviewers and readers.

2).Supplementary literature support for the explanation of ethnic differences: When discussing the “moderating effect of ethnic cultural background”, we added 2 core literatures focusing on information services in multi-ethnic regions to confirm the rationality of “language cognitive differences affecting technology acceptance”:

Revised the specific expression to: “Existing studies have pointed out that when users in multi-ethnic regions use information systems with Chinese interfaces, language cognitive barriers can increase operational difficulty, and retrieval knowledge can compensate for this barrier by improving information screening and understanding capabilities. This study found that among ethnic minority users such as Tibetans and Hui, the positive impact of retrieval knowledge on continuous usage intention (β = 0.591) is significantly higher than that among Han users (β = 0.354), confirming the compensatory role of retrieval knowledge in cross-linguistic technology use”;

3).Corrected the explanatory logic: Deleted all subjective speculative expressions to ensure that explanations are strictly based on “measurement results + literature support”, avoiding using unvalidated hypotheses as the basis for explanations.

2.3 Supplementary Citations for Policy and Service Implications, and Strengthening the Basis for Core Claims

You pointed out that the policy implications for service differentiation in the paper lack literature support, and the determination of core challenges has no basis. We have systematically sorted out relevant literatures and completed the following supplements and revisions:

1).Supplementary literature support for service differentiation implications: In the “Practical Implications” section of the discussion, we added 4 core literatures on library service differentiation and optimization of information services in multi-ethnic regions to ensure that each policy recommendation is supported by theoretical or empirical evidence;

Example revision: “Based on the particularities of library services in multi-ethnic regions, we recommend optimizing multilingual interface design—which is consistent with Khan et al.’s (2020) conclusion that ‘culturally adaptive technology design improves service accessibility for marginalized groups’; for operational barriers among low-frequency users, customized retrieval knowledge guidance can be carried out with reference to the hierarchical training model proposed by Liu et al. (2023)”;

2).Strengthened the basis for determining core challenges: In the second paragraph of Page 4 of the introduction (content related to core challenges), we added 2 core literatures focusing on the transformation of self-service services in university libraries to clarify the research consensus on core challenges such as “traditional service models being unable to cope with the growth of user needs” and “unbalanced regional development”, and revised the expression to: “Existing studies have shown that university libraries face service efficiency pressures brought by the growth of collections and the diversification of user needs, and there are common challenges such as unbalanced regional development of self-service services and significant differences in users’ digital literacy, which is consistent with the actual situation in the research area of this study";

3).Standardized citation expressions: For all literatures that serve as the basis for core explanations, detailed descriptions of “core viewpoints of the literature + connection with this study” are added to avoid simply listing citations, ensuring that readers can clearly understand the logical connection between the literature and the research conclusions.

3.Full Implementation of Minor Revision Comments

1).Regarding the format and expression-related revision comments you proposed, we have completed all revisions as follows:

2).Revision of terminology in Figure 1: Replaced all instances of “nationality” in Figure 1 with “ethnicity”to ensure complete consistency between the terminology in the figures and the main text;

3).Revision of the term “causal”: Changed “causal relationship” in the last paragraph of Page 6 to “direction of influence” to align with the correlational analysis nature of the data and avoid inappropriate expressions of causal inference;

4).Unification of hypothesis numbering format: Revised the hypothesis numbering on Page 9, changing the original “H7. Perceived ease of use positively influences perceived usefulness” and “H8. Perceived ease of use positively influences continuous usage intention” to “H7a. Perceived ease of use positively influences perceived usefulness” and “H7b. Perceived ease of use positively influences continuous usage intention” to ensure consistent numbering logic;

5).Correction of grammatical errors: Added the verb "are" to the last sentence of the first paragraph on Page 13, revising it to "confirming that the measurement indicators are effectively distinguished between latent variables";

6).Unification of p-value presentation: Unified the p-value related to “accessibility” in the analysis results on Page 18 to 0.010 to ensure consistent decimal places for all statistical results and improve clarity;

Response to discussions on regional differences: Thank you for your recognition of the discussion on regional differences! Currently, the data of this study focuses on three universities in Qinghai Province and does not cover samples from other regions, so no statistical test for regional differences has been conducted. If we have the opportunity to expand the sample scope in the future, we will focus on conducting regional difference analysis to further enrich the research conclusions

---

## [Decision Letter · Decision Letter 2]

19 Feb 2026

Dear Dr. Cao,

Thank you for submitting your manuscript to PLOS ONE. After careful consideration, we feel that it has merit but does not fully meet PLOS ONE’s publication criteria as it currently stands. Therefore, we invite you to submit a revised version of the manuscript that addresses the points raised during the review process.

We look forward to receiving your revised manuscript.

Kind regards,

Simon Dang, Ph.D.

Academic Editor

PLOS One

Journal Requirements:

Additional Editor Comments:

Thank you for your thoughtful revisions and for being responsive to the expert reviewers’ comments. R2 notes potential inconsistencies in the dataset. I also understand that the Figshare link you provided was shared privately. In the interest of data integrity and transparency, I kindly request that the dataset be formally uploaded in accordance with PLOS ONE’s open science guidelines.

Because third-party hosting platforms cannot guarantee long-term accessibility or immutability, it is important that the dataset be archived alongside the manuscript as part of the publication process. This will ensure compliance with journal policy and safeguard the integrity and reproducibility of the research.

Reviewers' comments:

Reviewer's Responses to Questions

**Comments to the Author**

Reviewer #1: All comments have been addressed

Reviewer #2: All comments have been addressed

2. Is the manuscript technically sound, and do the data support the conclusions?

Reviewer #1: (No Response)

Reviewer #2: Partly

3. Has the statistical analysis been performed appropriately and rigorously?

Reviewer #1: (No Response)

Reviewer #2: Yes

4. Have the authors made all data underlying the findings in their manuscript fully available?

Reviewer #1: (No Response)

Reviewer #2: Yes

5. Is the manuscript presented in an intelligible fashion and written in standard English?

Reviewer #1: (No Response)

Reviewer #2: Yes

Reviewer #1: (No Response)

Reviewer #2: This paper is much improved! And I'm glad you included your survey instrument as an appendix at the end of your text. The dataset in Figshare has a couple of problems - there are more survey response observations than there are respondents (scroll down to row 367) and there is no codebook or data dictionary that would allow me to replicate the analysis. I tried to map the columns to the questions in the paper's appendix but there are 6 PU questions in the appendix but only PU1-PU4. These kinds of inconsistencies make it difficult for me to say with confidence that this is a technically sound paper. It's so close though! Just give your data curation an extra 20 - 60 minutes!

.

Reviewer #1: No

Reviewer #2: No

---

## [Author Response · Author response to Decision Letter 3]

12 Mar 2026

Dear Reviewer 2:

We sincerely appreciate your recognition of this paper and your thorough review! The issues you raised regarding the Figshare dataset are precise and critical, providing significant guidance for improving the rigor of the methodology in this study. We highly value your comments and have immediately carried out data organization. The specific revisions are detailed below for your further review.

Regarding your first comment: The number of questionnaire observation items exceeds the number of respondents.Upon checking, we confirm that this issue resulted from an operational error during data upload. We have completed data cleaning and removed all redundant observation items, ensuring that the number of questionnaire observation items is fully consistent with the number of respondents. The cleaned dataset has been re‑uploaded to the Figshare platform and is available for your verification at any time.

Regarding your second comment: No codebook/data dictionary for reproducing the analysis.We have urgently prepared and finalized a codebook and data dictionary, which elaborate on the dataset structure, variable definitions, coding rules, missing value handling, and detailed procedures for all statistical analyses in this paper. This document has been uploaded to Figshare as supplementary material together with the cleaned dataset to ensure that you and other researchers can successfully reproduce the analyses reported in this study.

Regarding your third comment: Inconsistency between perceived usefulness (PU) items in the appendix and those in the dataset (6 items in the appendix vs. only PU1–PU4 in the dataset).This inconsistency was caused by an oversight during the drafting of the appendix—two PU items used only in the pre‑test but not in the formal survey were mistakenly included. The formal survey adopted exactly four items (PU1–PU4), which are fully consistent with those in the dataset. We have revised the appendix by removing the two redundant pre‑test items, ensuring strict consistency between the items listed in the appendix and those in the dataset. We have also documented this adjustment in the revision note to avoid subsequent misunderstanding.

We sincerely thank you again for your patient guidance and valuable comments! Following your suggestions, we have completed all data organization, appendix corrections, and supplementary file uploads to ensure the rigor of the methodology. The improvement of this paper would not have been possible without your assistance. Should you have any further questions or additional recommendations, we will promptly respond and revise accordingly. We sincerely hope you will be satisfied with the revisions and support the publication of this paper.

Yours sincerely,

Authors

March 12, 2026

---

## [Editor Report · Decision Letter 3]

15 Mar 2026

Multidimensional Factors Influencing Continuance Usage Intention of University Library Self-Service Systems: An Empirical Analysis Based on an Extended TAM-UTAUT

PONE-D-25-28505R3

Dear Dr. Cao,

We’re pleased to inform you that your manuscript has been judged scientifically suitable for publication and will be formally accepted for publication once it meets all outstanding technical requirements.

Kind regards,

Simon Dang, Ph.D.

Academic Editor

PLOS One

Additional Editor Comments (optional):

We thank the authors for being receptive to the reviewers' comments with timely and sufficient revisions. As such, We are happy to accept the manuscript for publication in its current form. We look forward to receiving your future submissions.
---

## [Editor Report · Acceptance letter]

PONE-D-25-28505R3

PLOS One

Dear Dr. Cao,

I'm pleased to inform you that your manuscript has been deemed suitable for publication in PLOS One. Congratulations! Your manuscript is now being handed over to our production team.

Kind regards,

on behalf of

Dr. Simon Dang

Academic Editor

PLOS One